# LIFT the Veil for the Truth: Principal Weights Emerge after Rank Reduction for Reasoning-Focused Supervised Fine-Tuning

**Zihang Liu** [1]  **Tianyu Pang** [2]  **Oleg Balabanov** [1 3 4]  **Chaoqun Yang** [5]  **Tianjin Huang** [6 7]
**Lu Yin** [8 7]  **Yaoqing Yang** [2]  **Shiwei Liu** [9 7]

## Abstract

Recent studies have shown that supervised fine-tuning of LLMs on a small number of high-quality datasets can yield strong reasoning capabilities. However, full fine-tuning (Full FT), while powerful, is computationally expensive and susceptible to overfitting and catastrophic forgetting, particularly when data is limited. Sparse fine-tuning, which previously achieved notable success by updating only a small subset of model parameters, offers a promising trade-off between efficiency and effectiveness. Yet, it has lagged behind in the LLM era due to the difficulty of identifying parameters truly critical for reasoning. In this work, we state that weights with the largest magnitude after low-rank approximation are critical weights for fine-tuning, which we call ***Principal Weights***. Surprisingly, while magnitude-based sparse fine-tuning performs poorly as a baseline on LLM fine-tuning, it becomes highly effective after rank reduction. These insights motivate our method: **L**ow-rank **I**nformed Sparse **F**ine-**T**uning (`LIFT`). `LIFT` only updates the top 5% *Principal Weights* throughout training and consistently achieves better performance on reasoning tasks than Full FT, while maintaining memory efficiency on par with popular parameter-efficient fine-tuning methods. In addition to strong performance on target domains such as arithmetic reasoning, `LIFT` also retains up to 20% more source-domain knowledge, compared to Full FT and LoRA. Our code is available at: https://github.com/zihanghliu/LIFT.

[1]University of California, Berkeley, CA, USA [2]Dartmouth College, NH, USA [3]International Computer Science Institute, CA, USA [4]Lawrence Berkeley National Laboratory, CA, USA [5]Tsinghua University, China [6]University of Exeter, Exeter, UK [7]Eindhoven University of Technology, the Netherlands [8]University of Surrey, Guildford, UK [9]University of Oxford, Oxford, UK. Correspondence to: Zihang Liu <zihang.liu@berkeley.edu>, Shiwei Liu <shiwei.liu@maths.ox.ac.uk>.

*Proceedings of the 42$^{nd}$ International Conference on Machine Learning*, Vancouver, Canada. PMLR 267, 2025. Copyright 2025 by the author(s).

## 1. Introduction

Large language models have recently undergone a revolutionary advancement in reasoning capabilities through Supervised Fine-Tuning (SFT) (Ye et al., 2025; Muennighoff et al., 2025) and Reinforcement Learning (RL) (Guo et al., 2025; Face, 2025). Performing SFT on a small, high-quality dataset delivers remarkable reasoning performance on math problems (Muennighoff et al., 2025). However, Full Fine-Tuning (Full FT) is prone to overfitting on limited training data (Chu et al., 2025) and incurs substantial computational costs due to the massive sizes of modern LLMs (Yin et al., 2023).

On the other hand, Sparse Fine-Tuning (Sparse FT) (Guo et al., 2020; Xu et al., 2021; Sung et al., 2021; Sanh et al., 2020a), a standout approach for pre-LLM fine-tuning, has demonstrated promising performance by training only a small subset of the base model's parameters. However, the adoption of Sparse FT has significantly lagged behind its low-rank counterparts in LLMs, as it struggles to identify parameters truly critical to fine-tuning, and its memory overhead is the same as Full FT with irregular sparse patterns.

In this paper, we propose **L**ow-rank **I**nformed Sparse **F**ine-**T**uning (`LIFT`), an effective and efficient approach for reasoning-focused LLM fine-tuning. `LIFT` builds on a counter-intuitive finding: the most naive baseline for Sparse FT, i.e., magnitude-based fine-tuning, becomes remarkably effective after applying low-rank approximation. We hence identify the weights with the largest magnitude after rank reduction as *Principal Weights*. The process of obtaining *Principal Weights* is illustrated in Figure 1. Empirical results show that `LIFT` outperforms state-of-the-art PEFT methods, Sparse FT methods, and Full FT on a wide range of tasks. `LIFT` **solves the challenges of Sparse FT** in these ways:

**Prior Knowledge:** `LIFT` identifies *Principal Weights*, which are critical for retaining pre-training knowledge and adapting to downstream tasks. The intuition aligns with recent findings that reasoning capacity is already in base models (Ye et al., 2025; Yue et al., 2025). `LIFT` further finds that this knowledge is encoded with *Principal Weights* and fine-tuning only these parameters is sufficient to achieve comparable—or even superior—reasoning performance.

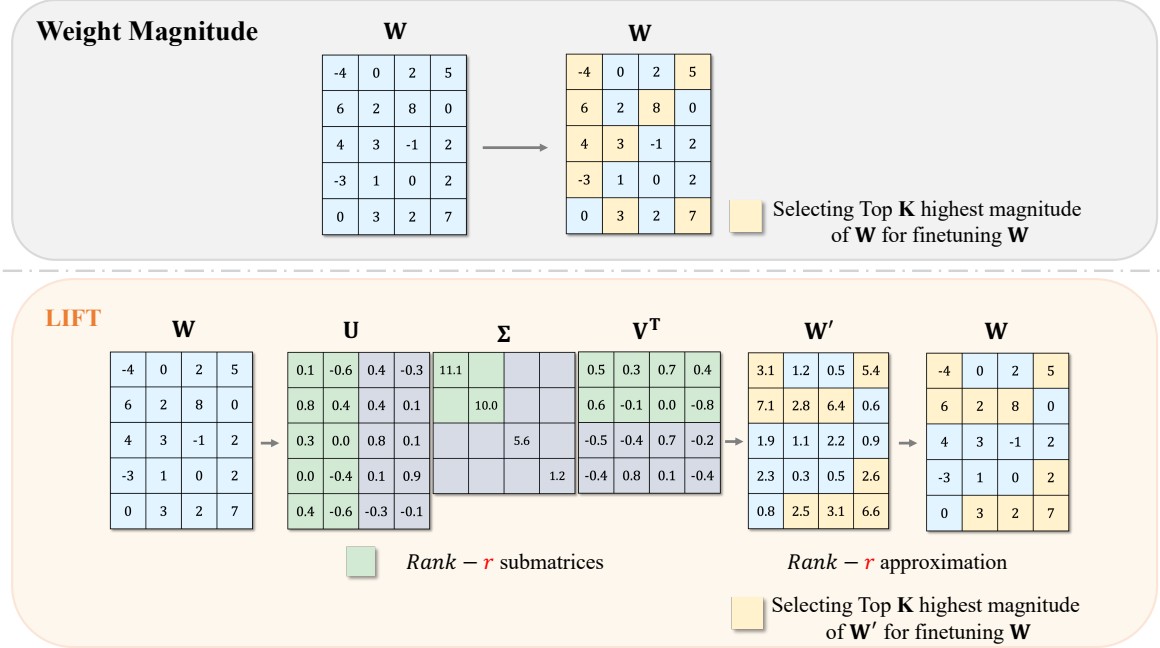

Figure 1: Overview of LIFT. LIFT first performs SVD on the original weight matrix $W$ to obtain a Rank-r approximation $W'$. It then selects the top-K parameters of $W'$ with the highest magnitudes to create a fine-tuning mask. This mask is then applied to the original weight matrix $W$ for fine-tuning.

**Memory Efficiency:** LIFT offers significantly better memory efficiency than Full FT and is on par with LoRA. LIFT updates and stores only a small subset of parameters during fine-tuning, leading to substantial memory savings—particularly in optimizer states, which are reduced from 27GB in Full FT to just 1.3GB ($<$5%) on LLaMA-2-7B.

Our analysis reveals that *Principal Weights* are more important for LLM fine-tuning compared to other weight selection criteria: Adding random perturbation to *Principal Weights* drastically affects model performance, substantially greater than other sparse selection metrics, both for pre-training knowledge and downstream tasks. In addition, the update matrix of LIFT has a substantially larger magnitude than LoRA and Full FT, and a substantially larger rank than that of LoRA, close to Full FT, enabling a larger capacity to acquire new knowledge in fine-tuning. Furthermore, LIFT can strongly affect the principal eigenspace of the LLM, making a significantly larger deviation than LoRA and Full FT, leading to better adaptation of the downstream tasks. We summarize our contributions as follows:

- We propose LIFT, a memory-efficient Sparse Fine-tuning algorithm, that selects and fine-tunes *Principal Weights*, as parameters with the highest magnitude after low-rank approximation. LIFT has significantly lower memory overhead than Full FT (less than 5% memory for optimizer), similar to that of LoRA. We show that

*Principal Weights* are crucial for retaining pre-trained knowledge and adapting to downstream tasks.

- We show that LIFT consistently yields strong performance on diverse sets of tasks. We evaluate LIFT on a wide range of reasoning benchmarks, including GPQA Diamond, Commonsense Reasoning, Arithmetic Reasoning, and Natural Language Understanding. We show that LIFT outperforms state-of-the-art PEFT methods, Full FT, and other Sparse FT strategies. Specifically, LIFT achieves up to 4.42% better performance than LoRA on commonsense reasoning, and up to 2.02% higher overall performance than Full FT on GPQA Diamond.

- We provide a comprehensive analysis of LIFT, and show that LIFT has **1) Strong generalization performance**, that it balances learning and forgetting, achieving stronger performance on target domains while up to 20% better than LoRA on source domains; **2) Strong learning capacity**, that it enables larger weight updates, with rank significantly higher than LoRA, close to Full FT, as well as other fascinating observations.

## 2. Related Works

**Low-rank Approximation of Weight Matrices.** Recent works (Sharma et al., 2024; Chen et al., 2025; Wang et al., 2024; Wei et al., 2024; Jaiswal et al., 2024; Geva et al., 2023)

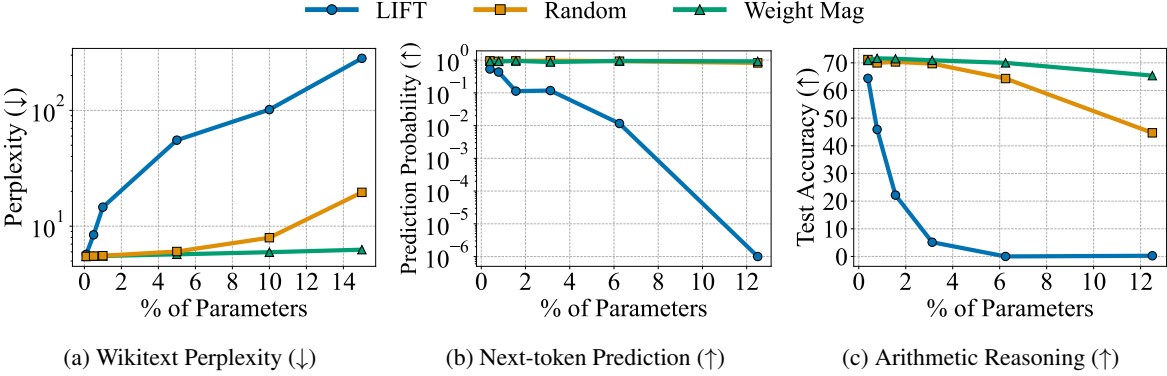

Figure 2: Evaluating pre-trained LLaMA-2-7B model with random noise added to selected parameters.

study the low-rank structure within the weight matrices of LLMs. Specifically, Sharma et al. (2024) and Chen et al. (2025) found that low-rank approximation of the feedforward layer can improve reasoning capabilities. Sharma et al. (2024) proposes that higher-order components of a weight matrix can introduce noise in decision-making, and eliminating the higher-order components "denoises" the model and helps recover "hidden", less-frequent information, improving the model's performance on questions whose answers are supported by less frequently occurring data. Chen et al. 2025 further theoretically confirmed this point in a two-layer transformer setting. The low-rank structures in weight update also inspire a line of adapter-based PEFT methods (Hu et al., 2022; Liu et al., 2024a; Meng et al., 2024; Huang et al., 2025).

**Eigenspectrum Analysis in Training and Fine-tuning.** Previous works (Ba et al., 2022; 2023; Wang et al., 2023) demonstrated theoretically that a two-layer neural network initialized with i.i.d. weights trained with one update step tends to exhibit a bulk+spike pattern in the empirical spectral density (i.e., the histogram of eigenvalues), which contains important signals in the corresponding eigenvectors. Recent works (Yang et al., 2023; Martin et al., 2020; Dandi et al., 2024) have investigated the eigenspectrum of weight/feature matrices of the trained model and found that shape metrics related to the heavy-tail distribution of eigenspectrum can reflect the quality of the model. Other works (Sanyal et al., 2019; Nassar et al., 2020; He & Ozay, 2022; Zhou et al., 2023; Liu et al., 2024b; Lu et al., 2024; Hu et al., 2025; Lingam et al., 2024; Meng et al., 2024) leveraged the information of the eigenspectrum and designed adaptive training and fine-tuning algorithms that improve generalization.

**Sparsity in Fine-tuning.** Sparse Fine-tuning (Sparse FT) aims to train a small subset of the weight matrices that are critical for downstream tasks, while achieving a smaller memory footprint than Full FT (Guo et al., 2020; Xu et al., 2021; Sung et al., 2021; Sanh et al., 2020a; Ansell et al., 2024; Song et al., 2024; He et al., 2025). Sparse FT achieved notable success before the era of modern LLM.

However, there is a lack of metrics that identify critical weights for fine-tuning, making it challenging to scale up. Li et al. (2024) prioritizes layers with significant outliers and fine-tunes models with sparse layers. Recently, Yang et al. (2024) proposed a structured Sparse FT scheme that achieves strong generalization performance. In this work, our proposed `LIFT` identifies the *Principal Weights* that are crucial to model performance.

## 3. Methodology

In this section, we introduce `LIFT` in detail. In Section 3.1, we describe the background and foundations for our approach. Section 3.2 presents the detailed algorithm of `LIFT`. An overview of `LIFT` is depicted in Figure 1.

### 3.1. Observations on Sparsity and Low-rank Structures

Sparse fine-tuning involves choosing a small subset of *Principal Weights* to fine-tune. Intuitively, the *Principal Weights* should be critical to model performance on different tasks. To determine the critical components in the weight matrices, recent works (Sharma et al., 2024; Chen et al., 2025) propose that lower-order components (corresponding to large singular values) contain information related to the task and context, while higher-order components (with smaller singular values) contain more generic information, which can be considered noise. These noise terms could prevent the adaptation process when adapting to a domain-specific task in fine-tuning. Based on this insight, we aim to design a sparse fine-tuning algorithm that finetunes parts of the weight matrices that contribute significantly to the model when the "noises" are filtered out through low-rank approximation. Essentially, the algorithm should finetune parameters with the largest magnitude after low-rank approximation.

### 3.2. `LIFT` Algorithm

We now introduce `LIFT` in detail. Given a model, for all trainable weight matrices $\{\mathbf{W}_1, \mathbf{W}_2, \ldots, \mathbf{W}_n\}$, `LIFT` first performs a $rank - R$ approximation of the weight matrices

| Model | Method | Best Rank | BoolQ | PIQA | SIQA | HellaSwag | Wino | ARC-e | ARC-c | OBQA | Avg. |
|---|---|---|---|---|---|---|---|---|---|---|---|
| | Full FT | – | 73.8 | 84.2 | 81.0 | **94.7** | 85.2 | 88.9 | 75.6 | 84.8 | 83.53 |
| | LoRA | 128 | 70.8 | 82.8 | 79.4 | 92.9 | 83.4 | 86.3 | 71.6 | 82.8 | 81.25 |
| | DoRA | 128 | 71.3 | 83.4 | 80.1 | 92.3 | 84.0 | 86.1 | 71.4 | 85.8 | 81.80 |
| LLaMA-2-7B | PiSSA | 128 | 72.5 | **85.3** | 80.8 | 87.2 | **86.1** | 87.1 | 74.3 | 85.6 | 82.36 |
| | S2FT | 128 | 73.3 | 83.7 | 81.0 | 94.3 | 84.6 | 88.3 | 75.8 | 84.8 | 83.22 |
| | LIFT | 128 | **74.8** | 84.7 | **82.2** | 94.4 | 86.0 | **89.2** | **76.4** | **89.6** | **84.66** |
| | Full FT | – | 75.4 | 88.0 | 81.8 | 96.5 | 89.3 | 93.1 | 83.0 | 86.0 | 86.64 |
| | LoRA | 64 | 71.8 | 85.3 | 80.9 | 93.4 | 84.5 | 90.0 | 77.0 | 84.8 | 83.46 |
| LLaMA-3-8B | DoRA | 64 | 74.6 | 87.4 | 81.2 | 94.7 | 87.1 | 89.4 | 79.5 | 86.4 | 85.04 |
| | S2FT | 64 | 67.7 | 89.8 | 82.5 | 95.2 | 87.8 | 93.1 | **84.6** | 88.6 | 86.16 |
| | LIFT | 32 | **75.7** | **90.5** | **83.2** | **96.5** | **89.4** | **93.6** | 83.9 | **90.2** | **87.88** |

Table 1: Commonsense reasoning. Fine-tuning on Commonsense-170K dataset.

to obtain $\{\mathbf{W}'_1, \mathbf{W}'_2, \ldots, \mathbf{W}'_n\}$, such that

$$\mathbf{W}'_i = \arg\min_{\text{rank}(\mathbf{W}'_i) \leq r} \|\mathbf{W}_i - \mathbf{W}'_i\|_F. \tag{1}$$

By obtaining the $Rank - r$ approximation, we filter out the "noisy" information in higher-order components, while maintaining the proximity to the original weight matrix according to the Eckart–Young–Mirsky theorem (Eckart & Young, 1936).

Then, LIFT generates a binary mask, in which positions with the highest magnitude are set to 1, and the rest to 0:

$$M_{ij} = \begin{cases} 1 & \text{if } W'_{ij} \text{ in top-}k \text{ of } W', \\ 0 & \text{otherwise.} \end{cases} \tag{2}$$

where k is the number of chosen parameters in $\mathbf{W}$. The mask is then applied to the original model during fine-tuning. Given any optimization algorithm with stochastic gradient updates, suppose at iteration $t$ the gradient of the weight matrix $W^t_i$ is $g^t_i$, we apply the binary mask $M^t_i$ to only store the gradient and corresponding optimizer states of the selected parameters as vectors:

$$g^t_i = \text{vec}(g^t_i[M^t_i = 1]) \tag{3}$$

By storing only the optimizer states of *Principal Weights*, the memory expenses of optimizers such as Adam are dramatically reduced. The detailed algorithm can be found in Appendix A. In Section 7.4, we show that the memory overhead of LIFT is significantly lower than Full FT, similar to that of LoRA.

In addition, as the low-rank approximation and its largest components will also change, we need to adjust our estimation of the *Principal Weights* dynamically. We choose the update interval that balances effectiveness and efficiency. A detailed discussion is in Appendix B.1.

## 4. LIFT Finds Principal Weights

We now try to provide initial insights into LIFT. We expect that through low-rank approximation, the model weights discard higher-order components (corresponding to smaller singular values) and retain the parts that best represent the encoded knowledge (Ba et al., 2022; Chen et al., 2025), leading to LIFT identifying the *Principal Weights*. To determine the importance of weights selected by LIFT, we design a simple experiment to randomly perturb different groups of weights and observe the resulting performance changes. If a set of parameters is truly critical to the LLM, perturbing them should cause a significant negative impact on the model performance. We empirically show that adding noise to parameters chosen by LIFT significantly affects model performance, compared to other selection criteria.

**Experiment Setup.** For an LLM, given a subset of parameters selected by a criterion, we add a Gaussian noise with a fixed scale of 0.01 to these parameters. Then, we evaluate the perturbed model on three tasks: **1) Wikitext Perplexity**, **2) Next-token Prediction**, and **3) Arithmetic Reasoning**. For 1) and 2), we use a pre-trained LLaMA-2-7B model to verify *Principal Weights*'s importance for **attaining pre-trained knowledge**, and for 3), we use a LLaMA-2-7B model fine-tuned on the MATH-10K dataset to verify the importance of *Principal Weights* on **fitting downstream tasks**. We compare the performance of the perturbed model under LIFT, and different parameter selection strategies.

**Wikitext Perplexity.** We first evaluate the perplexity of the perturbed model on the Wikitext dataset. As shown in Figure 2a, we see that when LIFT-selected parameters are added noise, the perplexity increases significantly, while other selection metrics remain stable. This implies that parameters selected by LIFT have a significant influence on the model's basic language capabilities.

**Next-token Prediction.** Recent work (Sharma et al., 2024; Chen et al., 2025) showed that replacing weight matrices with their low-rank approximation improves reasoning, predicting contextual answers instead of "generic" tokens as by

| Model | Method | Best Rank | MultiArith | GSM8K | AddSub | AQuA | SingleEQ | SVAMP | MAWPS | Avg. |
|---|---|---|---|---|---|---|---|---|---|---|
| LLaMA-3.2-1B | Full FT | – | **98.50** | **33.13** | **92.15** | 24.02 | 94.29 | 51.1 | **87.40** | 68.66 |
| | LoRA | 128 | 98.50 | 30.48 | 89.87 | 24.41 | 92.91 | 53.6 | 86.13 | 67.98 |
| | DoRA | 128 | 98.50 | 30.48 | 89.87 | 24.41 | 92.91 | 53.6 | 86.13 | 67.98 |
| | PiSSA | 128 | 97.33 | 32.15 | 91.90 | 23.23 | 94.09 | 52.3 | 86.97 | 68.28 |
| | S2FT | 128 | 96.17 | 30.17 | 90.38 | 23.62 | 92.13 | 49.4 | 81.93 | 66.26 |
| | LIFT | 128 | 98.17 | 32.37 | 90.89 | **26.38** | 92.91 | **56.3** | 86.13 | **69.02** |
| LLaMA-3.2-3B | Full FT | – | 97.83 | 56.71 | 92.66 | **33.46** | 95.08 | 69.1 | 90.34 | 76.45 |
| | LoRA | 128 | 98.50 | 55.95 | 91.39 | 27.95 | 94.29 | 70.3 | 91.60 | 75.71 |
| | DoRA | 128 | 98.33 | 55.12 | 91.65 | 28.35 | 94.88 | 70.9 | 89.92 | 75.59 |
| | PiSSA | 128 | 98.33 | **59.51** | 91.90 | 25.20 | 95.47 | 69.9 | 89.50 | 75.69 |
| | S2FT | 128 | 98.33 | 55.65 | 91.90 | 29.53 | 96.06 | 68.9 | 88.66 | 75.58 |
| | LIFT | 128 | **99.17** | 57.92 | **94.17** | 28.74 | **96.85** | **71.0** | 91.60 | **77.06** |
| LLaMA-2-7B | Full FT | – | 98.17 | 46.55 | **93.67** | 22.05 | 96.85 | 63.2 | 89.08 | 72.79 |
| | LoRA | 128 | 98.00 | 47.76 | 92.41 | 23.62 | 95.08 | 62.9 | **90.76** | 72.93 |
| | DoRA | 64 | 98.00 | 47.38 | 92.41 | 21.26 | 96.06 | 62.3 | 89.50 | 72.42 |
| | PiSSA | 128 | 98.83 | **48.45** | 92.66 | 21.26 | 95.87 | 63.4 | 90.76 | 73.03 |
| | S2FT | 128 | **99.17** | 44.43 | 91.39 | **29.13** | 95.47 | 62.6 | 89.50 | 73.10 |
| | LIFT | 128 | 98.67 | 47.31 | 92.66 | 26.77 | **96.85** | 63.6 | 90.34 | **73.74** |
| LLaMA-3-8B | Full FT | - | 99.00 | 69.83 | 93.42 | 28.74 | 97.83 | 79.6 | 92.86 | 80.18 |
| | LoRA | 64 | 99.17 | 71.57 | 92.15 | 24.41 | 96.26 | 80.5 | 92.02 | 79.44 |
| | DoRA | 64 | 98.83 | 70.96 | 90.89 | 29.53 | 96.65 | **81.8** | 90.76 | 79.92 |
| | PiSSA | 128 | 99.00 | 71.27 | **93.67** | 28.74 | 97.64 | 80.6 | 92.02 | 80.42 |
| | S2FT | 64 | **99.67** | 70.89 | 92.91 | 32.68 | 97.64 | 78.2 | **94.12** | 80.87 |
| | LIFT | 128 | 99.33 | **72.40** | 93.42 | **34.65** | **98.03** | 80.9 | 93.70 | **81.78** |

Table 2: Arithmetic reasoning. Fine-tuned on the MATH-10K dataset.

the original matrix. Inspired by this, we aim to investigate the role of parameters selected by `LIFT` on the next-token prediction task. Specifically, we use pre-trained LLaMA-2-7B model and analyze its output probability given a prompt such as "Madrid is located in the country of". Given this prompt sentence, the pre-trained model would successfully predict the ground truth answer "Spain". On the other hand, if the model fails to learn the context information, it is more likely to behave like an n-gram model and predict generic words like "the". As shown in Figure 2b, after more `LIFT`-selected weights are added noise, the output probability of "Spain" decays drastically, and eventually becomes zero. While other selection metrics are not influenced by the noise, still predict the correct answer.

**Arithmetic Reasoning.** Figure 2c shows the average test accuracy on 7 arithmetic reasoning tasks of the perturbed model. We can see that when parameters selected by `LIFT` are added noise, the performance degrades drastically to 0, compared to other metrics where adding noise doesn't have a significant influence.

The above experiments show that parameters selected by `LIFT` are extremely crucial to model performance, and they are sensitive to tiny perturbations. Therefore it makes sense to focus on these parameters during fine-tuning, as they will potentially be more adaptive to downstream tasks, and be more robust after fine-tuning. In Appendix C, we provide insights into `LIFT` from the perspective of spectral norm.

## 5. Experiments

In this section, we evaluate `LIFT` on various fine-tuning tasks, and compare it with state-of-the-art fine-tuning methods, including Full FT, LoRA (Hu et al., 2022), DoRA (Liu et al., 2024a), Spectral Adapters (Zhang & Pilanci, 2024), S2FT (Yang et al., 2024), and PiSSA (Meng et al., 2024). We also compare `LIFT` with sparse fine-tuning methods such as SpIEL (Ansell et al., 2024) and SIFT (Song et al., 2024). We demonstrate that `LIFT` achieves superior performance than these methods across all datasets. To ensure a fair comparison, we reproduce all results of the baseline methods with the codebases provided in previous papers.

### 5.1. Experimental Setup

**Tasks.** We conduct experiments on domains of major interests to modern LLM communities, including: **1) Reasoning Models**, in which we perform SFT with Qwen-2.5 on s1K dataset (Muennighoff et al., 2025) and evaluated on GPQA Diamond (Rein et al., 2023); **2) Commonsense Reasoning**, where we fine-tune models on Commonsense-170K dataset (Hu et al., 2023) and evaluate them on eight commonsense reasoning tasks; **3) Arithmetic Reasoning**, in which we fine-tune models on the MATH-10K dataset (Hu et al., 2023) and evaluate them on 7 arithmetic datasets; **4) Natural Language Understanding**, in which we fine-tune and evaluate models on the GLUE datasets (Wang et al., 2018); **5) Code generation**, where we perform instruct fine-tuning and evaluate the model on the Humaneval dataset;

| Method | Best Rank | MNLI | SST-2 | MRPC | CoLA | QNLI | QQP | RTE | STSB | Avg. |
|--------|-----------|------|-------|------|------|------|-----|-----|------|------|
| Full FT | – | 90.22 | 96.10 | 89.71 | 70.67 | 93.59 | 92.20 | 83.03 | 91.38 | 88.36 |
| LoRA | 64 | 89.92 | 95.87 | 89.95 | 68.35 | 92.88 | 90.62 | 81.95 | 90.81 | 87.57 |
| DoRA | 64 | 89.93 | 96.22 | 89.95 | 68.59 | 92.86 | 90.66 | 82.31 | 90.83 | 87.66 |
| Spectral | 64 | 89.89 | 96.22 | 88.73 | 69.65 | 93.45 | 91.21 | 82.67 | 90.92 | 87.84 |
| PiSSA | 128 | 90.12 | 95.87 | 89.46 | 68.48 | 93.48 | 91.72 | 81.95 | 91.10 | 87.77 |
| LIFT | 128 | **90.49** | **96.56** | **90.93** | **71.84** | **93.90** | **92.38** | **85.92** | **91.86** | **89.24** |

Table 3: Natural language understanding. Fine-tuning DeBERTa-v3 on GLUE datasets

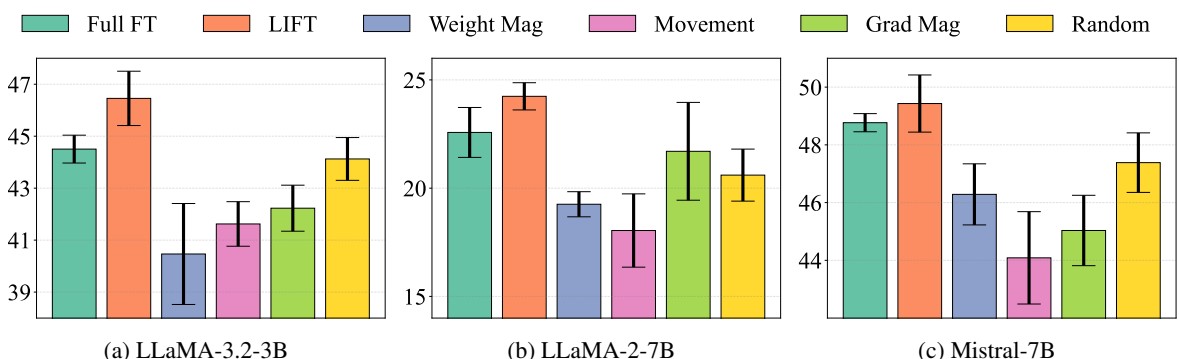

Figure 3: Comparing different sparse parameter selection metrics on GSM8K dataset.

**6) Question Answering**, where we fine-tune and evaluate models with the StrategyQA dataset.

**Models.** For arithmetic and commonsense reasoning, we use LLaMA models with sizes ranging from 1B to 8B (Touvron et al., 2023a;b; Grattafiori et al., 2024). For reasoning models with test-time scaling, we choose the instruction-finetuned Qwen-2.5 (Qwen et al., 2025) 1.5B and 3B models. For natural language understanding, we use DeBERTa-V3-base (He et al., 2023) and RoBERTa-large (Liu et al., 2019). For code generation, we use LLaMA-2-7B model. For question answering, we choose LLaMA-2-7B and LLaMA-3-8B.

To ensure fair comparison, we search the rank of PEFT methods in {16, 32, 64, 128, 256}, and LIFT with the same parameter counts, and compare their best results among all the ranks. For detailed dataset configurations and hyperparameter settings, please refer to Appendix D.

### 5.2. Reasoning Models on GPQA Diamond

Recently, reasoning models such as DeepSeek R1 (DeepSeek-AI, 2025), and Qwen 2.5 (Qwen et al., 2025) have shown advanced reasoning capabilities by scaling up compute resources during test-time. This trend incentivizes the development of more efficient and effective methods to train these reasoning models. To evaluate LIFT on adapting reasoning models, we follow the settings of the recent s1 paper (Muennighoff et al., 2025), and train instruct-finetuned Qwen-2.5 models with supervised fine-tuning on the s1K dataset.

| Metric | Qwen2.5-1.5B | Qwen2.5-3B |
|--------|--------------|------------|
| Full FT | 26.77 | 33.33 |
| LIFT | **28.79** ($r = 128$) | **34.85** ($r = 256$) |

Table 4: Test accuracies of Qwen 2.5 models on GPQA-Diamond, trained with supervised fine-tuning on the s1K dataset.

In Table 4, we compare LIFT with Full FT, which is the default method for supervised fine-tuning. We can see that LIFT can achieve better performance than Full FT on Qwen2.5 1B and 3B instruct model. This result shows the potential of LIFT on the training of large scale reasoning models.

### 5.3. Commonsense Reasoning

As shown in Table 1, LIFT achieves superior results on commonsense reasoning tasks than other fine-tuning methods. When compared to PEFT methods, LIFT outperforms DoRA and PiSSA by 2.86% and 2.30% with LLaMA-2-7B model; When compared to Full FT, LIFT achieves 1.24% and 1.13% higher overall accuracy with LLaMA-3-8B and LLaMA-2-7B respectively. This highlights the effectiveness of LIFT in commonsense reasoning.

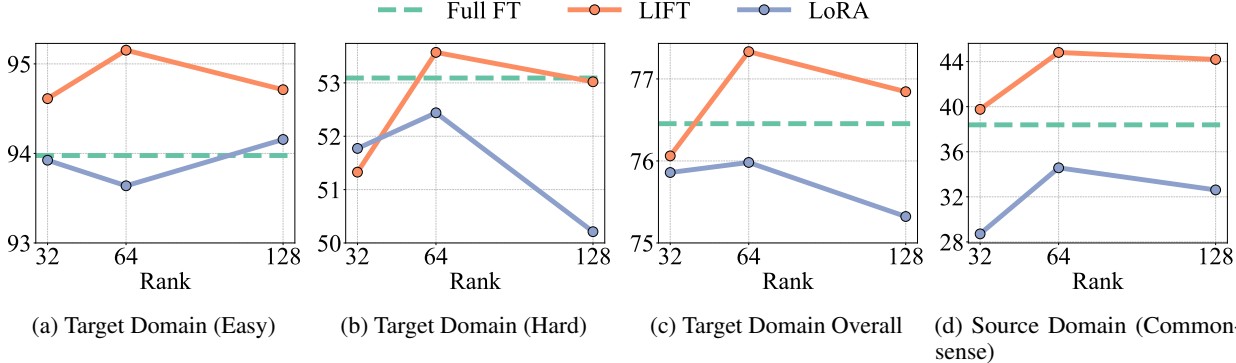

Figure 4: Generalization performance of LIFT on Near OOD and Far OOD tasks with LLaMA-3.2-3B.

### 5.4. Arithmetic Reasoning

In Table 2, we present evaluation results on seven arithmetic tasks. We can see that LIFT outperforms all PEFT methods, Sparse FT methods, and Full FT on LLaMA models. Specifically, compared to LoRA, LIFT achieves 1.79% higher overall performance on LLaMA-3.2-3B. In addition, LIFT outperforms the recent S2FT method by 2.76% on LLaMA-3.2-1B. More significantly, compared to Full FT, LIFT achieves a noticeable improvement of 1.14% and 1.60% on LLaMA-2.7B and LLaMA-3-8B, respectively. This suggests that LIFT can yield better test performance across a wide range of model sizes. Furthermore, LIFT achieves significantly better results on the most difficult tasks, such as GSM8K and SVAMP. This implies that LIFT can obtain high-level arithmetic capabilities more effectively.

### 5.5. Natural Language Understanding

We further evaluate the performance of LIFT on Natural Language Understanding tasks, as shown in Table 3. We can see that LIFT achieves the highest performance on every task, significantly better than other fine-tuning methods. Specifically, LIFT outperforms Full FT by 0.88% overall, and surpasses the recent Spectral Adapter method by 1.40%.

### 5.6. Additional Results

In addition to the results provided above, we have further evaluated LIFT on more diverse domains. This includes code generation under instruction fine-tuning, and question answering, which are presented in Appendix E.2 and Appendix E.3, respectively. We have also compared LIFT with other sparse fine-tuning methods, namely SpIEL (Ansell et al., 2024) and SIFT (Song et al., 2024), on diverse tasks, which are presented in Appendix F.

## 6. Ablation Study

In this section, we perform extensive ablation studies to support the effectiveness of LIFT. In Section 6.1, we compare LIFT with other sparse weight selection methods. Then in

Appendix B.1, we study the effect of update interval on the performance of LIFT. Finally in Appendix B.2, we compare the performance of different rank reduction strategies.

### 6.1. Comparing Different Parameter Selection Metrics

To evaluate the superiority of weight selection with LIFT, we compare LIFT with different parameter selection metrics on the task of fine-tuning on the GSM8K dataset. Specifically, we chose the pre-trained model of LLaMA-3.2-3B, LLaMA-2-7B, and Mistral-7B (Jiang et al., 2023), and we ran each experiment on four random seeds. The weight selection criteria include: 1) Weight Magnitude, 2) Movement Score (Sanh et al., 2020b), 3) Gradient Magnitude, and 4) Random Selection. As shown in Figure 3, we can see that LIFT outperforms all other parameter selection metrics by a large margin while surpassing Full Fine-tuning. This suggests that LIFT is a robust and effective selection metric for fine-tuning, especially on challenging tasks like GSM8K.

### 6.2. More Ablation Studies

Due to page limits, we place more ablation study results in Appendix B. In Appendix B.1 we study the effect of update interval of LIFT on model performance, and in Appendix B.2, we compare different rank reduction strategies.

## 7. Discussions

In this section, we provide comprehensive analysis on LIFT. First, in Section 7.1, we study how LIFT learns more on the target domain and forgets less on the source domain. In Section 7.2, we investigate the weight update of LIFT. Then in Section 7.3, we study the eigenspace and eigenspectrum of model fine-tuned by LIFT to investigate its learning dynamics. In Section 7.4, we analyze the memory efficiency of LIFT. Further discussion results are in Appendix G.

### 7.1. LIFT Balances Learning and Forgetting

Balancing learning and forgetting is crucial in studying the generalization abilities of a training algorithm. Follow-

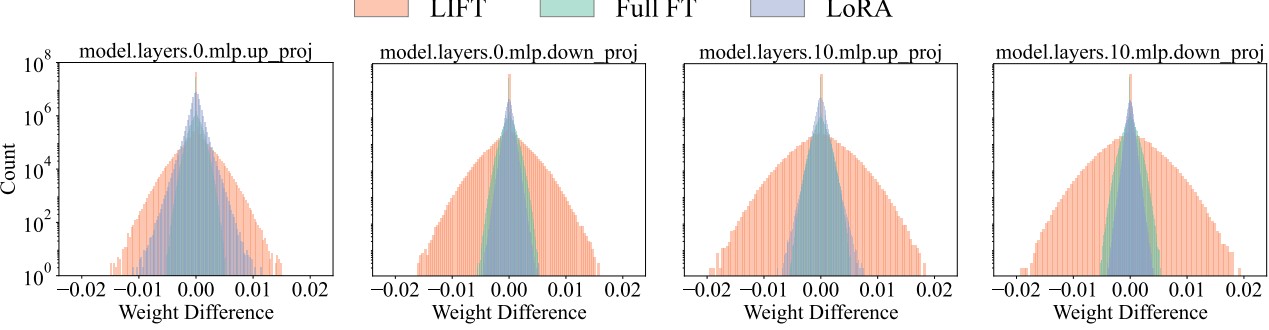

Figure 5: Weight difference between the model before and after fine-tuning.

ing (Biderman et al., 2024; Yang et al., 2024), we study the learning and forgetting of LIFT with pre-trained LLM fine-tuned on MATH-10K, and evaluating it on both arithmetic reasoning tasks (target domain), including easy (i.e, MultiArith, AddSub, SingleEq, MAWPS) and hard (i.e, GSM8K, AQuA, SVAMP), and commonsense reasoning tasks (source domain). Specifically, we compare the performance of LIFT with Full FT and LoRA. To ensure a fair comparison, we match the number of trainable parameters of LIFT and LoRA with the corresponding rank.

In Figure 4, we show the performance of LIFT with LLaMA-3.2-3B model. We can see that LIFT significantly outperforms Full FT and LoRA on both easy and hard tasks. In addition, LIFT also surpasses the source domain performance of Full FT and LoRA by a large margin, more than 5% than Full FT and 12% than LoRA. This suggests that LIFT is not only able to achieve superior results on the target domain tasks but also can retain previous knowledge, showcasing its ability to forget less. We believe that this strong generalization might come from the fact that LIFT only tunes parts of the weight matrices, keeping a large portion of the parameters unchanged, which makes the model forget less. In Figure 10 of Appendix G.1 we also show the performance of LIFT with LLaMA-3-8B model.

## 7.2. Weight Update

To analyze the changes brought by LIFT to the model, we plot the magnitude distribution of the weight update matrix $\Delta W$ of different layers in the model, as shown in Figure 5. We can see that LIFT brings a significantly larger weight update than Full FT and LoRA. These larger weight updates may reflect that the model is actively exploring and exploiting its capacity to capture new task-specific features, improving performance on the fine-tuning dataset. In the meantime, we can see that with LIFT, only a small set of parameters are changed while most weights remain unchanged (a large spike in the center of LIFT's magnitude distribution), the model retains its fundamental capacities that enable it to generalize to OOD settings.

## 7.3. Eigenspace and Eigenspectrum Analysis

To analyze how LIFT enables superior fine-tuning results, we investigate the dynamics of eigenspace and eigenspectrum before and after fine-tuning. Specifically, we compute the **1)** alignment score of top eigenspace (right singular vectors) before and after fine-tuning to measure the deviations in these directions, and **2)** rank of weight update. The definition of alignment score can be found in Appendix H.1

**Eigenspace.** In Figure 12, we show the alignment score of each layer of LLaMA-3.2-3B model (a larger alignment score means more similar eigenspace). We can see that:

- Some layers' eigenspace are extremely robust to fine-tuning. For example, for **Query**, **Key** and **Gate** layer, the alignment score is almost 1. This implies that fine-tuning these layers is less effective, as they don't bring further rotation to the top eigenspace. On the other hand, for **Output**, **Up**, and **Down** layers, the change of alignment scores is 10x larger than the other layers. This indicates that these layers are more adaptive to fine-tuning tasks, as fine-tuning them is more effective in rotating their eigenvectors. In Appendix G.2, we empirically demonstrate the difference in the effectiveness of fine-tuning these layers. We note that this observation is also observed in previous works (Sharma et al., 2024; Yang et al., 2024).

- LIFT is especially effective on highly adaptive layers. We can see that for **Output**, **Up** and **Down** layers, LIFT can bring significantly larger rotations to the top eigenspace, resulting in much lower alignment scores than Full FT and LoRA.

**Rank.** In Figure 13, we show the rank of the update matrix of each layer, grouped by their layer type. We can see that compared to LoRA, LIFT doesn't have rank constraints, and that the rank of the update is significantly higher than LoRA, close to Full FT. On some finetune-crucial layers such as Up and Down projection in the MLP module, LIFT achieves almost the same rank update as Full FT on all

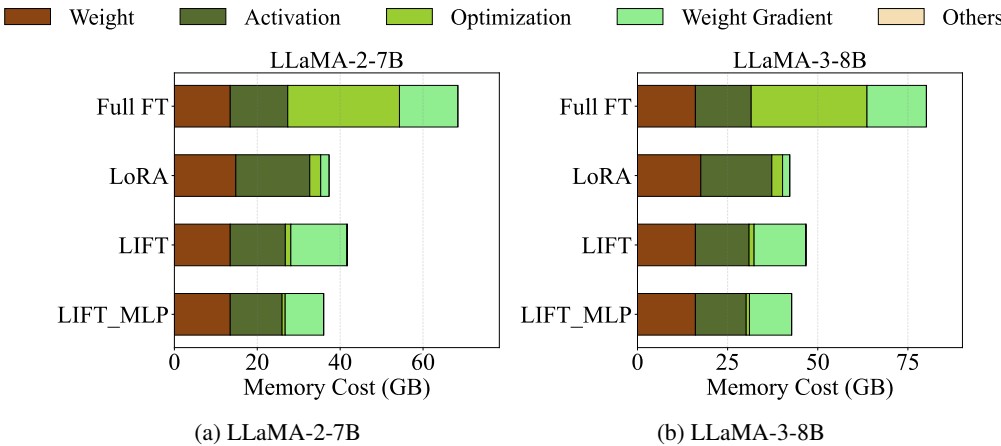

Figure 6: Breakdown of memory consumption of `LIFT`, LoRA and Full Fine-tuning.

layers. This suggests that `LIFT` has a larger capacity to learn task-related knowledge, which could explain its superior performance than other PEFT methods. Details of computing the rank are explained in Appendix G.3.

Combining the two metrics, we can see that `LIFT` serves as a good method to 1) provide larger eigenspace rotation to adapt to fine-tuning tasks, and 2) provide large rank update to increase the capacity of learned knowledge in fine-tuning.

### 7.4. Memory Efficiency of **`LIFT`**

We analyze the memory cost of `LIFT`. Figure 6 shows the memory overhead breakdown of `LIFT`, Full FT, and LoRA with LLaMA-2-7B and LLaMA-3-8B. We can see that the overall memory overhead is just slightly larger than that of LoRA, and significantly smaller than Full FT. Specifically, `LIFT` only takes up around 5% memory on the optimizer states than Full FT, due to the usage of sparse momentum and variance. This suggests that `LIFT` is able to effectively balance efficiency and performance. In Appendix G.4, we further show that the memory overhead of `LIFT` can be further reduced by only fine-tuning MLP layers, while achieving comparable performance (i.e. `LIFT_MLP`).

### 7.5. More Discussions

We provide further discussions on `LIFT` in Appendix G. In Appendix G.5, we create a two-layer model on a simple regression task to simulate the fine-tuning with `LIFT`, and show that `LIFT` has stronger generalization abilities than Full FT. In Appendix G.6, we analyze the training loss curve of `LIFT`. In Appendix G.7, we explore the potential of `LIFT` for structured sparse fine-tuning. In Appendix G.8, we analyze the influence of different ranks for rank reduction in `LIFT`. In Appendix G.9, we investigate the pattern of weights selected by `LIFT` compared to other methods.

## 8. Conclusion

In this paper, we propose that parameters with large magnitude after low-rank approximation of weight matrices are *Principal Weights* for reasoning-focused fine-tuning. Based on this insight, we designed a memory-efficient sparse fine-tuning algorithm `LIFT`, that fine-tunes only *Principal Weights*. We show that `LIFT` has superior performance than state-of-the-art PEFT methods, Full Fine-tuning, and Sparse FT methods on reasoning tasks. From extensive analysis, we find that: 1) `LIFT` learns more on the target domain while forgetting less in the source domain; 2) the weight update matrix of `LIFT` has a large magnitude and rank, providing a large capacity to adapt to fine-tuning tasks; and 3) `LIFT` brings large rotation to the top eigenspace of important layers. We hope that our work can provide insights into how to find critical weights in fine-tuning LLMs. In addition, we find that there are a few limitations that bring room for further exploration:

- How to combine `LIFT` with RL algorithms like GRPO to enhance the reasoning capacity of LLMs with better memory efficiency?

- How does the eigenvector rotation phenomenon of `LIFT` connect to the learning dynamics of LLM fine-tuning?

- Can `LIFT` be improved with GPU acceleration to further improve computation efficiency?

- Currently `LIFT` uses a global rank to perform LRA. However, different layers have different capacities. Can we improve `LIFT` by designing adaptive rank reduction on each layer?

We hope that these problems can inspire future research on the development of more effective fine-tuning approaches.

## Impact Statement

This paper presents work whose goal is to advance the field of Machine Learning and Deep Learning, especially the fine-tuning of Large Language Models. There are many potential societal consequences of our work, none of which we feel must be specifically highlighted here.

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

# A. `LIFT` Algorithm Detail

---

**Algorithm 1** Adam with `LIFT`

---

1: **Input:** $\theta_0, \alpha, \beta_1, \beta_2 \in [0, 1), \epsilon > 0, \texttt{update\_mask\_interval}, T$
2: $m_0 \leftarrow \mathbf{0}, v_0 \leftarrow \mathbf{0}, t \leftarrow 0, M \leftarrow \texttt{calc\_mask}(\theta_0)$
3: $m_{0\_update} \leftarrow \text{vec}(\mathbf{0}), v_{0\_update} \leftarrow \text{vec}(\mathbf{0})$
4: **for** $t = 1, 2, \ldots, T$ **do**
5:    **if** $t \mod \texttt{update\_mask\_interval} \equiv 0$ **then**
6:       $M_{old} \leftarrow M, M \leftarrow \texttt{calc\_mask}(\theta_t)\{\texttt{Update binary mask}\}$
7:       $m_{t-1}, v_{t-1} \leftarrow \mathbf{0}, \mathbf{0}$
8:       $m_{t-1}[M_{old} = 1] = m_{t-1\_update}$
9:       $v_{t-1}[M_{old} = 1] = v_{t-1\_update}$
10:      $m_{t-1\_update} = \text{vec}(m_{t-1}[M = 1])$
11:      $v_{t-1\_update} = \text{vec}(v_{t-1}[M = 1])$
12:    **end if**
13:    $g_t \leftarrow \nabla_\theta f_t(\theta_{t-1})$
14:    $g_{t\_update} = \text{vec}(g_t[M = 1])$
15:    $m_{t\_update} \leftarrow \beta_1 m_{t-1\_update} + (1 - \beta_1)g_{t\_update}$
16:    $v_{t\_update} \leftarrow \beta_2 v_{t-1\_update} + (1 - \beta_2)g_{t\_update}^2$
17:    $\hat{m}_{t\_update} \leftarrow \frac{m_{t\_update}}{1-\beta_1^t}, \hat{v}_{t\_update} \leftarrow \frac{v_{t\_update}}{1-\beta_2^t}$
18:    $\theta_t \leftarrow \theta_{t-1}, \text{vec}(\theta_t[M = 1]) \leftarrow \text{vec}(\theta_t[M = 1]) - \alpha\frac{\hat{m}_{t\_update}}{\sqrt{\hat{v}_{t\_update}}+\epsilon}$
19: **end for**

---

**`LIFT` as an adapter method.** We note that `LIFT` selects a fixed number of parameters to fine-tune at any training step. However, this does not guarantee `LIFT` has a fixed overall sparsity – the total count of parameters updated across the entire process can still fluctuate. Nevertheless, empirical evidence suggests that the total update matrix remains sparse (Figure 5).

To obtain a fixed-size adapter like in LoRA, one can construct the update mask in an accumulative manner, by gradually adding new principal weights to already-chosen parameters until a designated sparsity. Additionally, we can pre-determine the principal weights and fix them during fine-tuning. Such new versions of `LIFT` can have better portability and adaptability, and represent a promising direction for future research.

## B. More Ablation Studies

### B.1. Analyzing the update interval of `LIFT`

`LIFT` uses a dynamic scheme to update the selected parameters. To investigate the impact of update interval on model performance, we compare different update intervals, ranging from 50 to 1000, and compare the performance of LLaMA-2-7B model on the GSM8K dataset. Figure 7a shows the result. We can see that different intervals all significantly outperform the baseline, indicating the robustness of `LIFT`. Moreover, the interval should neither be too small nor too large, and a median interval is the best choice. This aligns with empirical insight as a smaller interval would change the fine-tuned weights more frequently, making some weights not fully trained, and from the perspective of compute efficiency, it would be also efficient if we don't update the mask too frequently. On the other hand, not changing the selected weights is also inferior as some weights may be already saturated in terms of training quality, and the low-rank approximation of the weight matrix also changes during training, therefore the *Principal Weights* also changes. Therefore, choosing an update interval that is not too small and not too large can best benefit the performance of `LIFT`.

### B.2. Comparing Different Rank Reduction Strategies

To prove that rank reduction by preserving the largest singular values and singular vectors (or ranks) is the best choice, we compare it with other rank reduction strategies, including randomly preserving the rank (Random), preserving the smallest rank (Smallest), and preserving a combination of largest and smallest ranks (Hybrid). The results are shown in Figure 7b. Specifically, we choose the setting of fine-tuning LLaMA-2-7B model on the MATH-10K dataset and compare the average

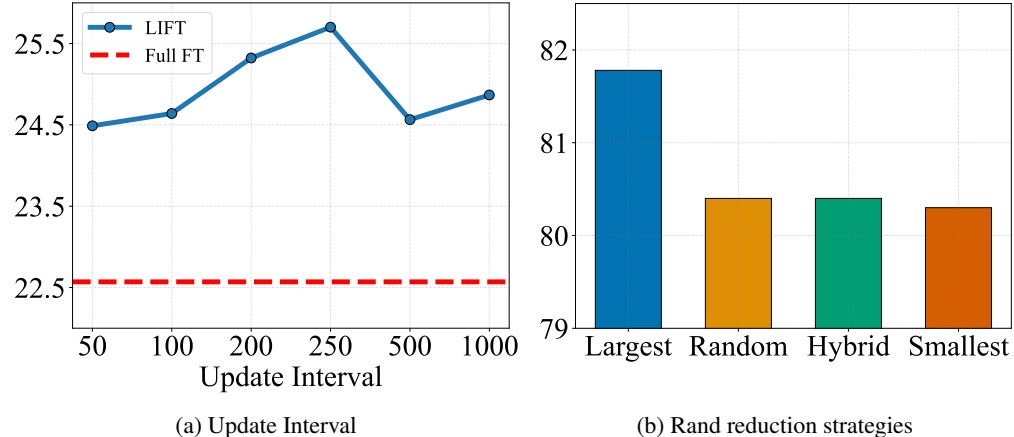

(a) Update Interval          (b) Rand reduction strategies

Figure 7: More ablation results. **(a)** Comparing GSM8K accuracy of different update interval of sparse mask chosen by `LIFT`. **(b)** Comparing different rank reduction methods on 7 arithmetic reasoning tasks (mean accuracy).

performance on 7 tasks. We can see that `LIFT`, by preserving the largest ranks, yields significantly better performance than other rank reduction strategies.

## C. More Intuitions on `LIFT`

In this section, we further analyze `LIFT` from a spectral norm perspective. The spectral norm of a matrix represents its largest singular value, which is the largest "stretch" of the transformation by the weight matrix to input data. Several theoretical works have also linked spectral norm to the generalization performance of the model (Neyshabur et al., 2015; Bartlett et al., 2017; Neyshabur et al., 2018; Galanti et al., 2023). Here we show that the update brought by `LIFT` can dramatically influence the spectral norm of the weight matrix, compared to other sparse update methods. We first analyze the influence of `LIFT` on a random matrix. Then, we investigate the change of spectral norm corresponding to Section 4.

### C.1. Random Matrix Case

To begin, we consider adding noise to `LIFT` selected by `LIFT` on a random matrix with different dimensions. Figure 8 shows the results of the spectral norm and Frobenius norm of the matrix before and after adding the noise, with different weight selection strategies. We can see that all methods don't differ from the Frobenius norm perspective. However, when it comes to spectral norm, while random selection and selection by weight magnitude doesn't change the spectral norm much, `LIFT` significantly increases the spectral norm, especially when the matrix size is large.

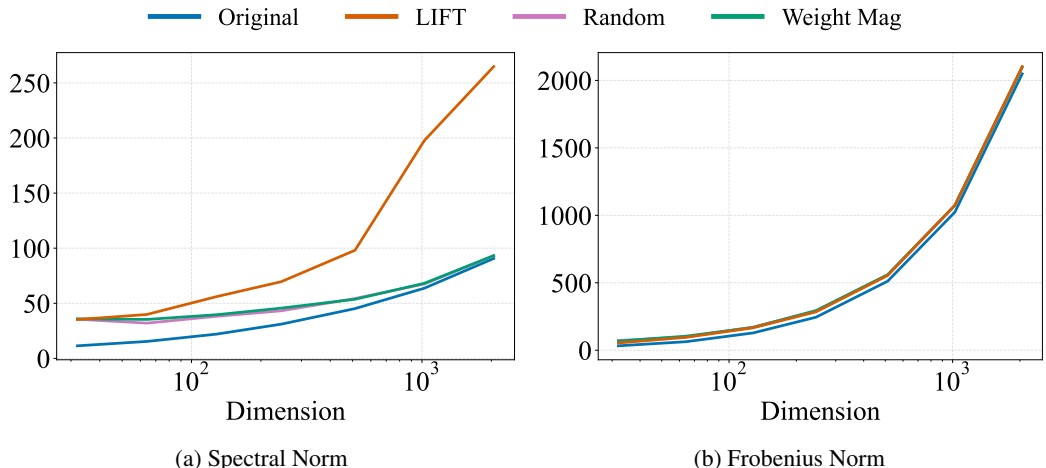

(a) Spectral Norm          (b) Frobenius Norm

Figure 8: Spectral Norm and Frobenius Norm of random matrices of different dimensions after adding random noise to selected weights.

## C.2. Adding Noise to LLM

Following the experiments in Section 4, we analyze the spectral norm of pre-trained LLM after adding random noise to weights selected by `LIFT`, compared to other sparse selection strategies. In Figure 9, we show the difference in spectral norm before and after adding the random noise with fixed scale of 0.1 to the LLaMA-2-7B model.

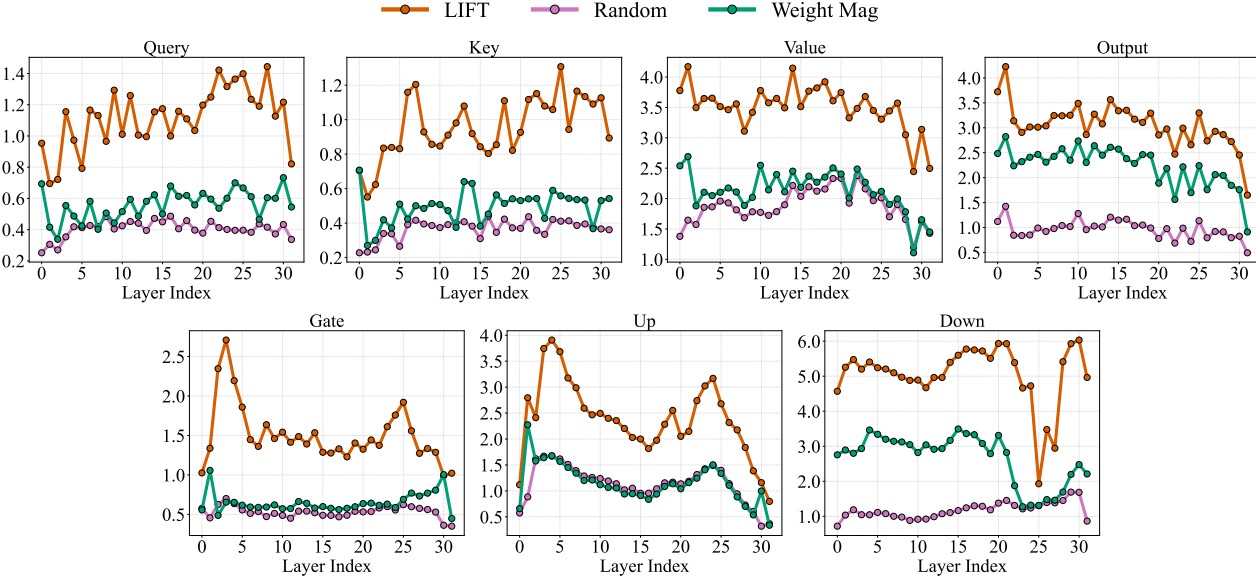

Figure 9: Spectral norm difference between before and after adding random noise to selected weights on pre-trained LLaMA-2-7B model.

We can see that `LIFT` brings a significantly larger change to spectral norm than other selection strategies. This indicates that the effect of `LIFT` on the spectral norm of weight matrices is consistent not only in toy settings as in Appendix C.1, but also in practical settings of LLMs.

# D. Detailed Experimental Setup

In this section, we provide a detailed experimental setup of `LIFT` of the main results. First, in Appendix D.1 we describe the detailed hyperparameters of experiments in Section 5. Then,

## D.1. Detailed Hyperparameters

The following tables corresponds to the main results in the paper. Table 5 corresponds to the experiments shown in Table 1; Table 6 corresponds to the experiments in Table 2 and 11; Table 7 corresponds to experiments in Table 3.

| Hyperparameters (**LIFT**) | LLAMA-2-7B | LLAMA-3-8B |
|---|---|---|
| Optimizer | AdamW | |
| LR | 1e-4 | 2e-4 |
| LR Scheduler | Linear | |
| Batch size | 16 | |
| Warmup Ratio | 0.03 | |
| Epochs | 3 | |

Table 5: Hyperparameter configurations for different LLaMA models on Commonsense-170K dataset. Corresponding to experiments in Table 1.

| Hyperparameters (LIFT) | LLAMA-3.2-1B | LLAMA-3.2-3B | LLAMA-7B | LLAMA-2-7B | LLAMA-3-8B |
|---|---|---|---|---|---|
| Optimizer | | | AdamW | | |
| LR | 2e-4 | 1e-4 | 2e-4 | 1e-4 | 2e-4 |
| LR Scheduler | | | Linear | | |
| Batch size | | | 16 | | |
| Warmup Ratio | | | 0.03 | | |
| Epochs | | | 3 | | |

Table 6: Hyperparameter configurations for different LLaMA models on MATH-10K dataset. Corresponding to experiments in Table 2 and 11.

| Hyperparameters (LIFT) | MNLI | SST-2 | MRPC | CoLA | QNLI | QQP | RTE | STSB |
|---|---|---|---|---|---|---|---|---|
| Optimizer | | | | AdamW | | | | |
| Epochs | 1 | 5 | 13 | 8 | 1 | 2 | 10 | 30 |
| LR | | | | {6e-5, 1e-4, 2e-4, 5e-4} | | | | |
| LR Scheduler | | | | Linear | | | | |
| Batch Size | | | | 32 | | | | |

Table 7: Hyperparameter configurations for different LLaMA models on GLUE datasets. Corresponding to experiments in Table 3.

## D.2. Searching Process for Number of Trainable Parameters

As shown in Section 5, we report the best result of each method with number of training parameters equivalent to LoRA rank in range {16, 32, 64, 128, 256}. Here we present the rank search process and provide detailed results of the performance of different methods under all ranks. Specifically, Table 8 is the rank search result of experiments in Table 1; Table 9 is the rank search result of experiments in Table 2; Table 10 is the rank search result of experiments in Table 3.

| LoRA Rank | 16 | 32 | 64 | 128 | 256 |
|---|---|---|---|---|---|
| Full FT | 86.64 | 86.64 | 86.64 | 86.64 | 86.64 |
| LoRA | 76.23 | 81.86 | **83.46** | 83.21 | 82.63 |
| S2FT | 81.07 | 85.03 | **86.16** | 83.82 | 82.76 |
| LIFT | 86.93 | **87.88** | 87.18 | 84.16 | 83.67 |

Table 8: Mean test performance on 8 commonsense reasoning tasks of different methods on LLaMA-3-8B with number of trainable parameters equivalent to various LoRA ranks. Bold numbers are the best test results of different methods. Corresponding to experiments in Table 1.

| LoRA Rank | 16 | 32 | 64 | 128 | 256 |
|---|---|---|---|---|---|
| Full FT | 72.79 | 72.79 | 72.79 | 72.79 | 72.79 |
| S2FT | 67.78 | 71.78 | 72.48 | **73.10** | 72.63 |
| PiSSA | 71.57 | 71.82 | 72.54 | **73.03** | 72.54 |
| DoRA | 71.10 | 71.74 | **72.42** | 71.83 | 71.81 |
| LoRA | 70.91 | 71.74 | 72.81 | **72.93** | 72.24 |
| LIFT | 70.91 | 71.09 | 72.74 | **73.74** | 73.67 |

Table 9: Mean test performance on 7 arithmetic reasoning tasks of different methods on LLaMA-2-7B with number of trainable parameters equivalent to various LoRA ranks. Bold numbers are the best test results of different methods. Corresponding to experiments in Table 2.

| LoRA Rank | 16 | 32 | 64 | 128 | 256 |
|---|---|---|---|---|---|
| Full FT | 88.36 | 88.36 | 88.36 | 88.36 | 88.36 |
| LoRA | 87.45 | 87.51 | **87.57** | 87.43 | 87.32 |
| DoRA | 87.35 | 87.56 | **87.66** | 87.45 | 87.41 |
| PiSSA | 87.13 | 87.21 | 87.40 | **87.77** | 87.33 |
| Spectral | 87.76 | 87.79 | **87.84** | 87.51 | 87.38 |
| LIFT | 85.17 | 86.93 | 88.19 | **89.24** | 88.74 |

Table 10: Mean test performance on GLUE tasks of different methods on DeBERTa-v3-base with number of trainable parameters equivalent to various LoRA ranks. Bold numbers are the best test results of different methods. Corresponding to experiments in Table 3.

## E. More Experimental Results

### E.1. Arithmetic Reasoning

To complement the arithmetic reasoning results in Figure 11, we show the test performance of LIFT on another model – LLaMA-7B, compared with the same set of fine-tuning algorithms.

| Model | Method | Best Rank | MultiArith | GSM8K | AddSub | AQuA | SingleEQ | SVAMP | MAWPS | Avg. |
|---|---|---|---|---|---|---|---|---|---|---|
| | Full FT | – | **98.50** | 42.68 | 91.90 | 22.83 | 95.08 | 59.5 | **89.50** | 71.43 |
| | S2FT | 128 | 98.50 | 40.49 | 92.15 | 24.80 | 95.28 | 57.2 | **89.50** | 71.13 |
| | PiSSA | 128 | 98.17 | 41.77 | **92.66** | **25.98** | **96.26** | 59.5 | 87.81 | 71.74 |
| LLaMA-7B | LoRA | 64 | 97.67 | **43.29** | 90.38 | 22.83 | 95.08 | 60.8 | 84.87 | 70.70 |
| | DoRA | 64 | 98.17 | 43.59 | 91.90 | 23.23 | 95.08 | 60.1 | 86.13 | 71.17 |
| | LIFT | 256 | 98.17 | 42.15 | 92.41 | 23.62 | 95.47 | **62.7** | 87.82 | **71.76** |

Table 11: Arithmatic reasoning performance on LLaMA-7B model. Fine-tuned on the MATH-10K dataset.

### E.2. Code Generation

To test the code generation performance of LIFT, we adopt the settings from the recent SIFT paper (Song et al., 2024), where we fine-tune LLaMA-2-7B with the Alpaca dataset for one epoch, and evaluate on the Humaneval dataset. From the table below, we can see that LIFT outperforms all other methods, in both pass@1 and pass@10 settings.

| | LIFT | Full FT | SIFT | LoRA | DoRA |
|---|---|---|---|---|---|
| Pass@1 | **16.46** | 15.24 | 14.02 | 13.66 | 13.96 |
| Pass@10 | **31.10** | 28.05 | 30.48 | 27.44 | 29.88 |

Table 12: Code generation performance of LIFT and different PEFT methods evaluated on the Humaneval dataset (↑), after training with the Alpaca dataset.

### E.3. Question Answering

To evaluate the performance of LIFT on question-answering tasks, we adopt the experimental setup of the StrategyQA dataset following recent WeLore paper (Jaiswal et al., 2024). For PEFT methods, we consider ranks {16, 32, 64, 128, 256}; for LIFT, we use the same counts of trainable parameters. The learning rates are {1e-5, 2e-5, 5e-5, 1e-4} for Full FT and {5e-5, 1e-4, 2e-4, 5e-4} for others. We select the best-performing config for each method and report the results below. We can see that LIFT achieves notable performance gains than all other methods, on both LLaMA-2-7B and LLaMA-3-8B model.

| | LIFT | Full FT | LoRA | DoRA | PiSSA |
|---|---|---|---|---|---|
| LLaMA-2-7B | **72.53** | 70.61 | 71.78 | 71.98 | 71.26 |
| LLaMA-3-8B | **75.85** | 74.81 | 74.44 | 74.27 | 75.19 |

Table 13: Question answering performance (↑) of LIFT and different PEFT methods evaluated on the StrategyQA dataset.

## F. Comparison with Other Sparse Fine-tuning Methods

In this section, we further support the empirical success of LIFT by comparing with recent sparse fine-tuning algorithms, including SpIEL (Ansell et al., 2024) and SIFT (Song et al., 2024).

### F.1. Comparing **LIFT** with SpIEL

We compare LIFT with SpIEL on the GSM8K dataset. We use the same training setting as Figure 3 in our paper. For both methods, we searched learning rate among {5e-5, 1e-4, 2e-4, 5e-4} and trainable parameters same as LoRA rank among {16, 32, 64, 128, 256} to obtain the best results. The table below shows that LIFT significantly outperforms SpIEL with both LLaMA-2-7B and LLaMA-3.2-3B.

| | LIFT | SpIEL | Full FT |
|---|---|---|---|
| LLaMA-3.2-3B | **46.46** | 43.76 | 44.50 |
| LLaMA-3-8B | **24.24** | 21.61 | 22.57 |

Table 14: Test accuracies on GSM8K dataset of LIFT, SpIEL, and Full FT.

### F.2. Comparing **LIFT** with SIFT

We compare LIFT with SIFT with two experimental setups following the SIFT paper. We use 1) Instruct tuning, where we fine-tune LLaMA-2-7B with the Alpaca dataset for one epoch, and evaluate on the Humaneval dataset, which is discussed in Appendix E.2, and 2) Natural Language Understanding, where we fine-tune the RoBERTa-large model on GLUE datasets. For both methods, we use the same number of trainable parameters (5% total parameters). We use the RoBERTa-large model, and search the learning rate of LIFT SIFT in {5e-5, 7e-5, 1e-4, 2e-4} and compare their best results. The table below shows that LIFT outperforms SIFT on all GLUE tasks, while outperforming Full FT on almost all tasks.

| Method | # Param (%) | MNLI | SST-2 | MRPC | CoLA | QNLI | QQP | RTE | STSB | Avg. |
|---|---|---|---|---|---|---|---|---|---|---|
| Full FT | – | 90.58 | 96.22 | **91.91** | 68.55 | 94.47 | 91.52 | 85.92 | 92.21 | 88.92 |
| SIFT | 5% | 89.91 | **96.79** | 89.95 | 66.29 | 93.04 | 88.49 | 87.00 | 92.27 | 87.97 |
| LIFT | 5% | **90.79** | 96.67 | 90.93 | **70.44** | **94.69** | **92.38** | **87.00** | **92.58** | **89.44** |

Table 15: GLUE results comparing LIFT, SIFT, and Full FT using RoBERTa-large.

## G. Complementary Discussions

### G.1. More Generalization Results

In addition to the results shown in Figure 4, we present the generalization results on the target and source domain of LLaMA-3-8B model in Figure 10. We can see that LIFT still achieves significantly stronger generalization results than Full Fine-tuning and LoRA fine-tuning, especially in hard target domain tasks (Figure 10b). In addition, we can see in Figure 10d that LIFT is able to generalize to source domain extremely well, achieving more than 10% better result than Full Fine-tuning, and almost 20% better than LoRA.

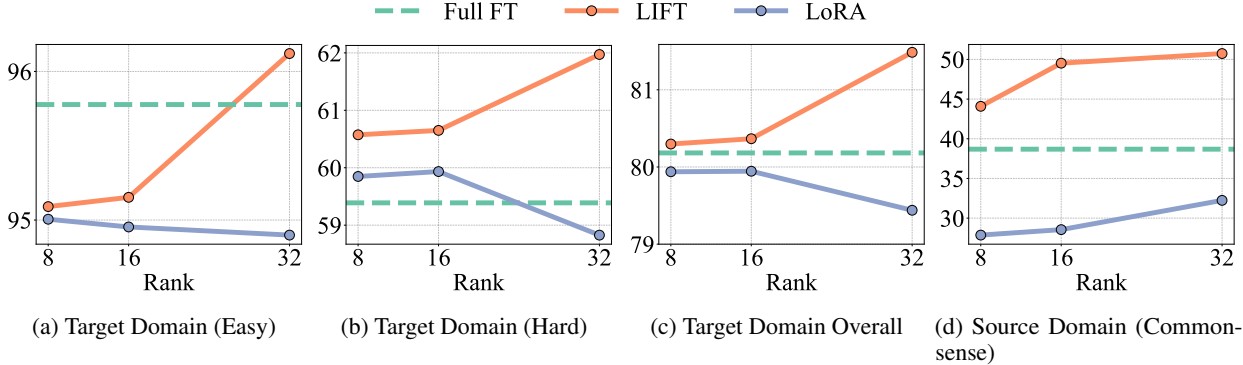

Figure 10: Performance of LIFT on target domain (arithmetic reasoning) and source domain (commonsense reasoning)

## G.2. Model Component Analysis

Following Section 7.3, we see that fine-tuning some layers is ineffective in adapting to downstream datasets. Therefore, we further study the effectiveness of LIFT of fine-tuning different model components. Specifically, in each experiment, we only fine-tune one layer type of the model, chosen from {Key, Query, Value, Output, Gate, Up, Down}. We evaluate the performance on 7 arithmetic reasoning tasks with LLaMA-2-7B model. The results are shown in Figure 11.

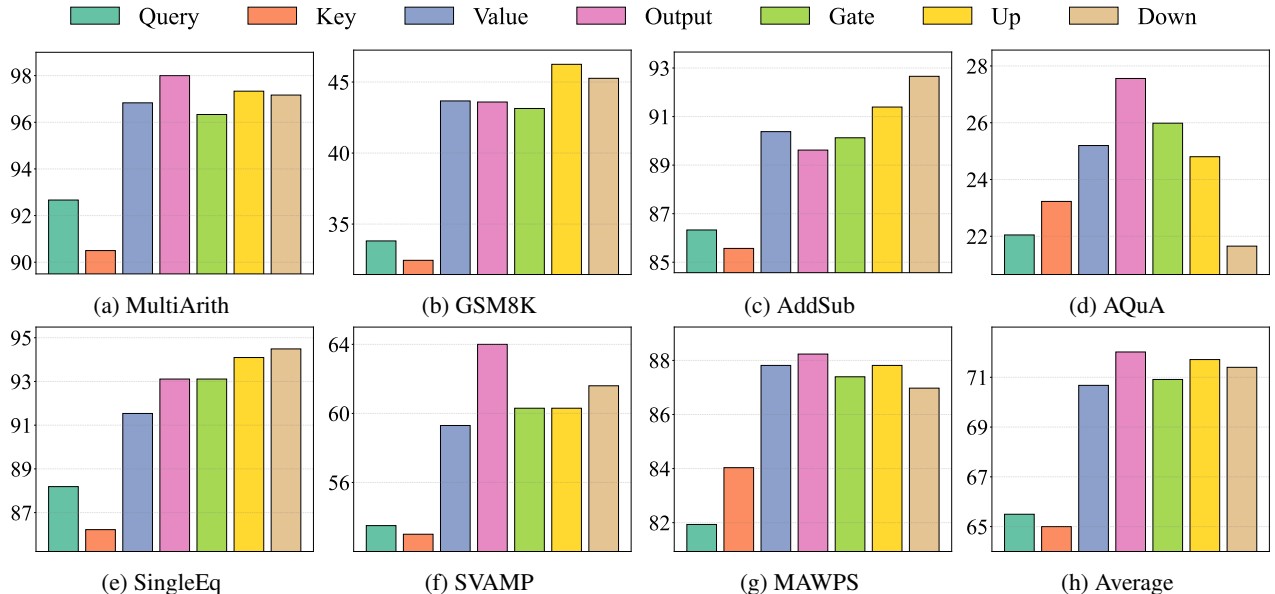

Figure 11: Average performance on 7 arithmetic reasoning tasks when applying LIFT with low-rank approximation of different ranks.

We can see that compared to other modules, fine-tuning only the **Query** and **Key** module yields significantly worse results on all tasks. In contrast, fine-tuning modules such as **Value**, **Up**, and **Down** modules achieves strong performance. This phenomenon is similar to the observation on the alignment of top eigenspace in Section 7.3, where the top eigenspace of Query and Key modules does not change much during fine-tuning, while Output, Up, and Down modules change relatively more significant. These observations could suggest that Query and Key modules are not very adaptable to downstream tasks, or that fine-tuning them may not be effective. This could be because the Key and Query components, as part of the self-attention module, mostly store information related to token relations, rather than task-specific knowledge. On the other hand, modules including the Output, Up, and Down projection layers mostly in the MLP module, tend to be more adaptive in fine-tuning, and is able to store more task-specific knowledge.

## G.3. Eigenspace and Eigenspectrum Analysis

Figure 12 and 13 shows the results corresponding to the discussion in Section 7.3. Figure 12 presents the alignment score between the top eigenspace before and after fine-tuning, comparing `LIFT`, LoRA and Full Fine-tuning. Figure 13 shows the rank of the weight update matrix of `LIFT`, Full Fine-tuning and LoRA.

**Rank computation.** To compute the ranks of different methods in Figure 13, we use `torch.linalg.matrix_rank` function from PyTorch. It counts the number of singular values greater than a threshold $\tau$, which has the default value:

$$\tau = \max(m, n) \times \sigma_{\max} \times \epsilon$$

where $(m, n)$ is the matrix shape, $\sigma_{\max}$ is the largest singular value, and $\epsilon$ is precision of input data type. Since robust rank comparison requires the threshold $\tau$ to exceed the rounding error incurred during the update matrix evaluation, we set it to **10 times the default value**.

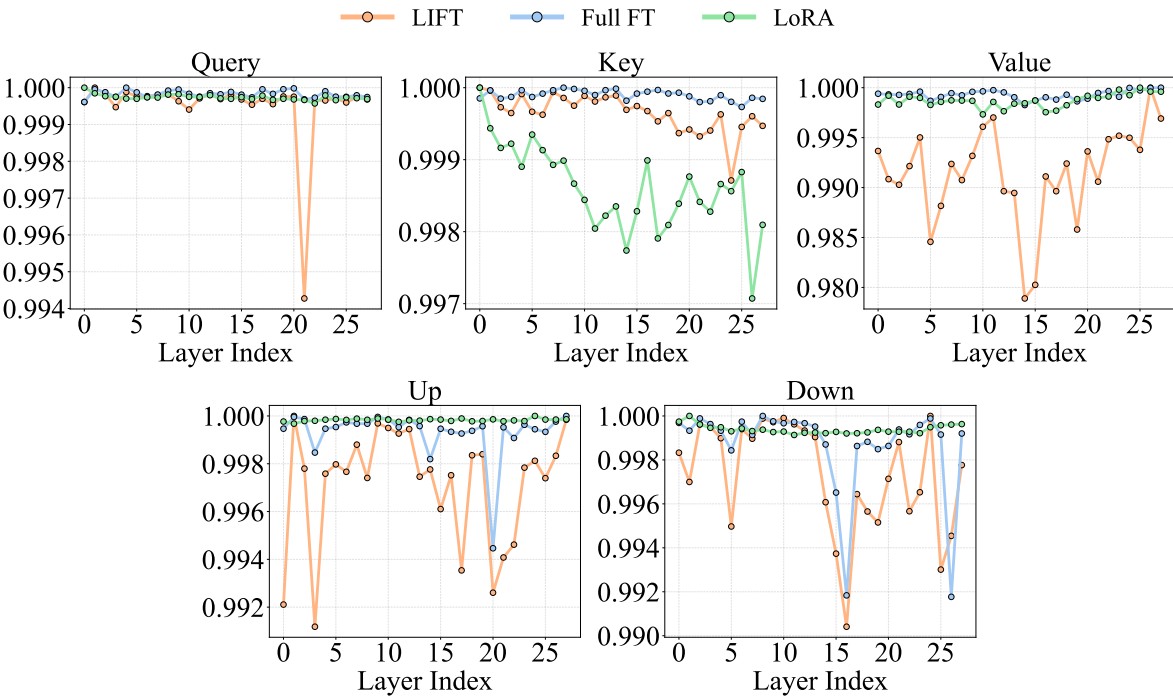

Figure 12: Alignment score of the top eigenspace of the weight matrix before and after fine-tuning on LLaMA-3.2-3B model. A lower alignment score indicates a larger deviation from the original eigenspace.

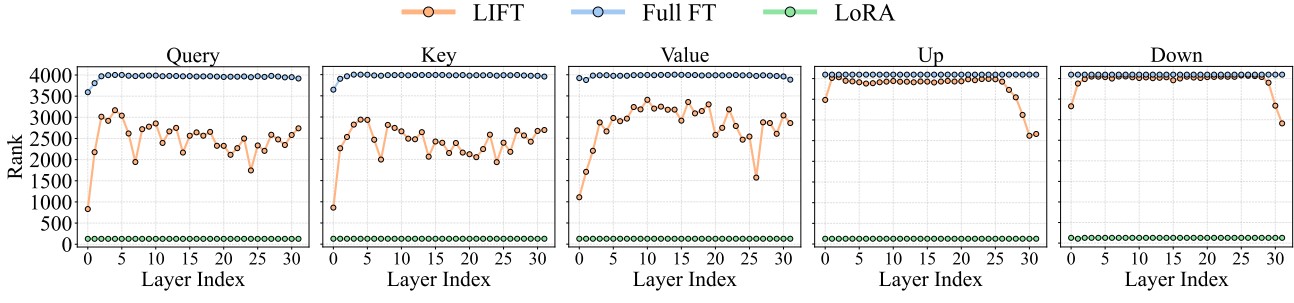

Figure 13: Rank of the weight update matrix of each layer of the LLaMA-2-7B model trained on MATH-10K dataset.

## G.4. Further memory saving of `LIFT`

In Appendix G.2, we show that fine-tuning MLP layers is more effective than fine-tuning attention layers. Table 16 shows the results of LLaMA-2-7B on arithmetic datasets where we **only fine-tune the MLP layers**, `LIFT_MLP`. We can see that

`LIFT_MLP` has similar performance to `LIFT`. From Figure 6, we can see that `LIFT_MLP` further reduces the memory usage on gradients and optimizer states, which achieves better memory efficiency than LoRA under optimal rank settings. We realize that selecting appropriate layers for `LIFT` fine-tuning is a crucial problem for future research, as it provides insights on the importance of different layers during fine-tuning, and can potentially further reduce the memory overhead.

| Method | Rank | MultiArith | GSM8K | AddSub | AQuA | SingleEQ | SVAMP | MAWPS | Avg. |
|---|---|---|---|---|---|---|---|---|---|
| LIFT | 128 | 98.67 | 47.31 | 92.66 | **26.77** | **96.85** | **63.60** | **90.34** | **73.74** |
| LIFT_MLP | 128 | **99.66** | **47.61** | 91.90 | 25.59 | 95.67 | 62.60 | **90.34** | 73.34 |
| Full FT | – | 98.17 | 46.55 | **93.67** | 22.05 | **96.85** | 63.20 | 89.08 | 72.79 |
| LoRA | 128 | 98.00 | 47.76 | 92.41 | 23.62 | 95.08 | 62.90 | 90.76 | 72.93 |

Table 16: Arithmetic reasoning performance of `LIFT_MLP`, which only fine-tunes the MLP layers of an LLM.

### G.5. A Toy Model Case

To demonstrate that `LIFT` works not only on large models, here we consider a toy model task: a two-layer neural network $f(x)$ for regression:

$$f(\mathbf{X}) = \sigma(\mathbf{X}\mathbf{W})\mathbf{a} \tag{4}$$

where $\mathbf{X} \in \mathbb{R}^{n \times d}, \mathbf{W} \in \mathbb{R}^{d \times h}, \mathbf{a} \in \mathbb{R}^{h \times 1}$.

**Training Pipeline.** We first pre-train this toy network with a curated training dataset then Apply `LIFT` during fine-tuning and compare its performance with that of Full Fine-tuning and other Sparse Fine-tuning methods. We consider `AdamW` optimizer and an early stopping strategy for training and fine-tuning.

**Pre-training and fine-tuning Datasets.** For pre-training dataset, we first random sample $\mathbf{X_{pre}} \in \mathbb{R}^{n_{pre} \times d}$, where $n_{pre}$ is the number of pre-training samples and $d$ is the input dimension. We construct pre-training labels

$$\mathbf{Y_{pre}} = \mathbf{X_{pre}}[:, : 32].sum(dim = 1) + 0.1 * \sin(\mathbf{X_{pre}}[:, 32 : 64]).sum(dim = 1) \tag{5}$$

For fine-tuning dataset, we random sample $\mathbf{X_{ft}} \in \mathbb{R}^{n_{ft} \times d}$, where $n_{ft}$ is the number of fine-tuning samples and $d$ is the input dimension. We construct fine-tuning labels.

$$\mathbf{Y_{ft}} = 0.2 * \mathbf{X_{ft}}[:, : 64] * \mathbf{X_{ft}}[:, : 65] * \mathbf{X_{ft}}[:, : 66] + 0.1 * \sin(\mathbf{X_{ft}}[:, 67] * \mathbf{X_{ft}}[:, 68]) \tag{6}$$

For convenince, we assume $d = 512, h = 128, n_{pre} = 5000, n_{ft} = 100$.

Figure 14 shows the statistics during fine-tuning of different methods. We compare `LIFT` with Full Fine-tuning and Sparse Fine-tuning with parameters selected by weight magnitude and gradient magnitude. As we can see, in terms of training and validation loss in Figure 14a and 14b, Full Fine-tuning saturates the early stage, indicating overfitting. On the other hand, sparse fine-tuning methods are more adaptive and achieve better generalization, achieving lower validation loss. Among the sparse fine-tuning methods, `LIFT` outperforms other strategies by a large margin, suggesting that `LIFT` has superior generalization performance. In addition, as shown in Figure 14c `LIFT` achieves a more stable gradient norm, with sharp decay towards the end, and converges to lower values. Furthermore, in Figure 14d, we can see that `LIFT` achieves a substantially lower spectral norm than all other methods. This observation aligns with Appendix C, that `LIFT` can significantly change the spectral norm of weight matrices. These observations on the effectiveness of `LIFT` pave the way for future theoretical works.

### G.6. Training Loss of `LIFT`

In Figure 15, we show the training loss curve of all methods of LLaMA-2-7B model from the experiments in Table 2. We can see that the convergence speed of LIFT is on par with Full FT, notably faster than other PEFT methods.

### G.7. Exploring Structured Fine-tuning of `LIFT`

In our paper, `LIFT` selects and fine-tunes model parameters in an unstructured fashion. We realize that `LIFT` naturally lends itself to block-sparse fine-tuning. To validate the great potential of `LIFT` in this context, we conducted an experiment with `LIFT_Structured` that selects and fine-tunes *Principal Weights* in $4 \times 4$ blocks. We compare it with two common

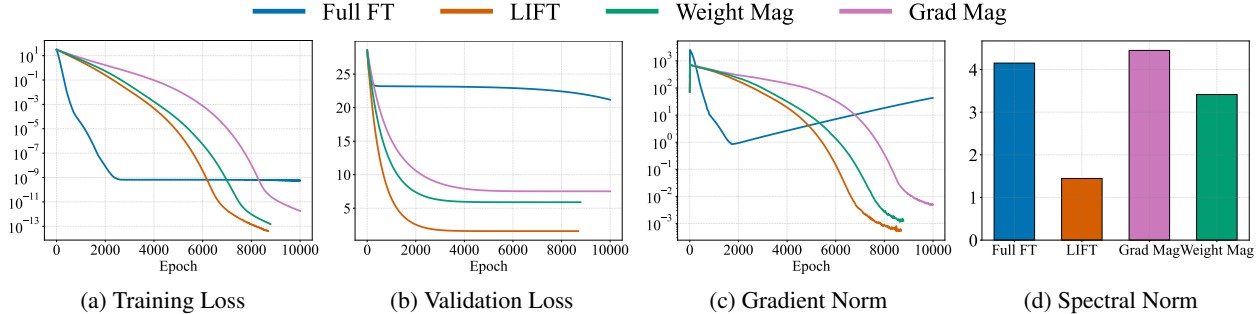

Figure 14: Fine-tuning statistics on the toy model setting of different fine-tuning strategies.

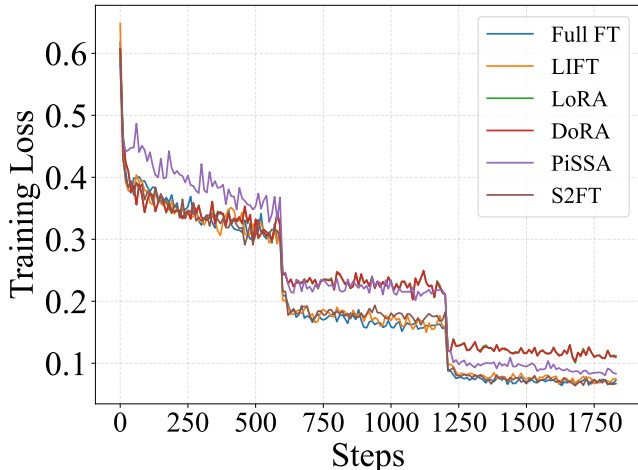

Figure 15: Training loss curves of different methods with LLaMA-2-7B model on MATH-10K dataset.

parameter selection metrics: 1) Top-k Magnitude, 2) Top-k Gradient. For all sparse methods, we fine-tune a subset of weights with the number of parameters identical to LoRA with rank $\approx 128$. We choose to fine-tune the LLaMA-2-7B model with the MATH10K dataset and evaluate on seven arithmetic reasoning tasks. The results are in Table 17.

| Method | Rank | MultiArith | GSM8K | AddSub | AQuA | SingleEQ | SVAMP | MAWPS | Avg. |
|---|---|---|---|---|---|---|---|---|---|
| LIFT_Structured | 128 | 98.33 | 48.07 | 93.16 | 25.98 | 95.47 | **65.1** | 89.92 | **73.72** |
| LIFT | 128 | **98.67** | 47.31 | 92.66 | **26.77** | **96.85** | 63.6 | **90.34** | **73.74** |
| Full FT | – | 98.17 | 46.55 | **93.67** | 22.05 | **96.85** | 63.2 | 89.08 | 72.79 |
| Weight Mag | 128 | 98.00 | **49.13** | 91.39 | 23.62 | 93.90 | 63.3 | 89.08 | 72.64 |
| Grad Mag | 128 | 97.50 | 45.72 | 92.41 | 23.23 | 95.28 | 60.0 | 88.66 | 71.83 |

Table 17: Performance of different parameter selection strategies on 7 arithmetic reasoning tasks.

We can see that LIFT_Structured still achieves great performance under structured sparsity. LIFT_Structured achieves performance on par with LIFT, and outperforms Full FT and other common sparse selection metrics. This suggests that LIFT has the potential to be adapted for structured sparse fine-tuning, enabling further computational acceleration.

## G.8. Rank Reduction of LIFT

In LIFT, we employ a uniform rank to perform low-rank approximation on each weight matrix. We know that models with different sizes possess different learning capacities, and their "Important Ranks" could potentially be different. To find the trend of how approximation rank influences the power of LIFT, we fine-tune models with different sizes using different approximation rank, ranging from 8 to 256.

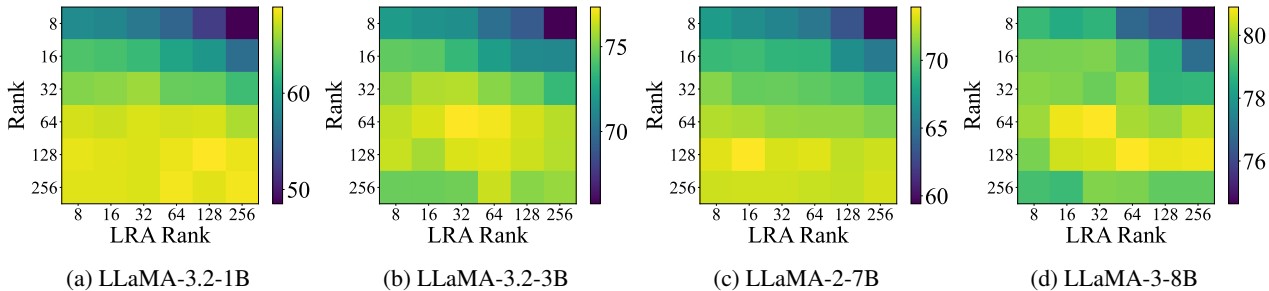

(a) LLaMA-3.2-1B      (b) LLaMA-3.2-3B      (c) LLaMA-2-7B      (d) LLaMA-3-8B

Figure 16: Average performance on 7 arithmetic reasoning tasks when applying LIFT with low-rank approximation of different ranks.

Figure 16 shows the heat map of different models on seven arithmetic reasoning tasks. In each plot, the x-axis represents the rank used for low-rank approximation, which we call the LRA rank. The y-axis rank represents the number of trained parameters, matching the number of parameters selected by the LoRA method with corresponding ranks, which we call Selected Rank. We can see that a larger LRA rank doesn't yield better performance. Instead, LIFT works best when the LRA rank is close to the Selected Rank. Moreover, when the model size is larger, the Selected Rank of best performance tends to be smaller than that of smaller models. This suggests that smaller models store knowledge more densely, therefore maintaining a larger effective rank, while the larger model has a larger capacity, and the effective rank for each layer may be smaller to store the same amount of knowledge.

### G.9. Parameter Selection of LIFT

To analyze the parameters chosen by LIFT, we compare them with parameters selected by Weight Magnitude, and analyze their overlap.

In Figure 17, we present the overlap of parameters selected by LIFT and Weight Magnitude on different layers. We can see that the overlap between the two methods is small for all layers. Specifically, for Query and Key layers, the overlap is relatively larger, generally around 20% and at most 40%, while on the Up and Down layers in the MLP module, the overlap is significantly smaller, around 5% and at most 20%. This suggests that the low-rank approximation of the Query and Key matrices are close to the original matrix, as the weights with the largest magnitude are more similar. This could mean that Query and Key modules are low-rank in nature, and fine-tuning them is not as effective as other modules. This aligns with our observation from Section 7.3 and Appendix 11.

In addition, when selecting the same amount of parameters, if we increase the rank of low-rank approximation, the overlap between LIFT and Weight Magnitude also becomes larger. This is also intuitive as we perform a more precise approximation of the original matrix as we increase the LRA rank.

## H. Metric Definitions

### H.1. Eigenspace Alignment Score

To calculate the alignment score of the top eigenspace before and after fine-tuning, we take the top 128 right singular vectors of each weight matrix before and after fine-tuning. For each singular vector $\mathbf{v}'_i$ after fine-tuning, we calculate the alignment score $d_i$ as the sum of squared cosine similarity with all top vectors of the pre-trained model $\mathbf{v}_1, \mathbf{v}_2, \ldots, \mathbf{v}_n$:

$$d_i = \sum_{j=1}^{n} (\mathbf{v}'_i \cdot \mathbf{v}_j)^2, \tag{7}$$

Since all singular vectors are orthonormal, the alignment score has the range $[0, 1]$, representing the projection length of the fine-tuned singular vectors on the subspace spanned by the top pre-trained singular vectors. Then, the total alignment score is the mean of the alignment score of all singular vectors,

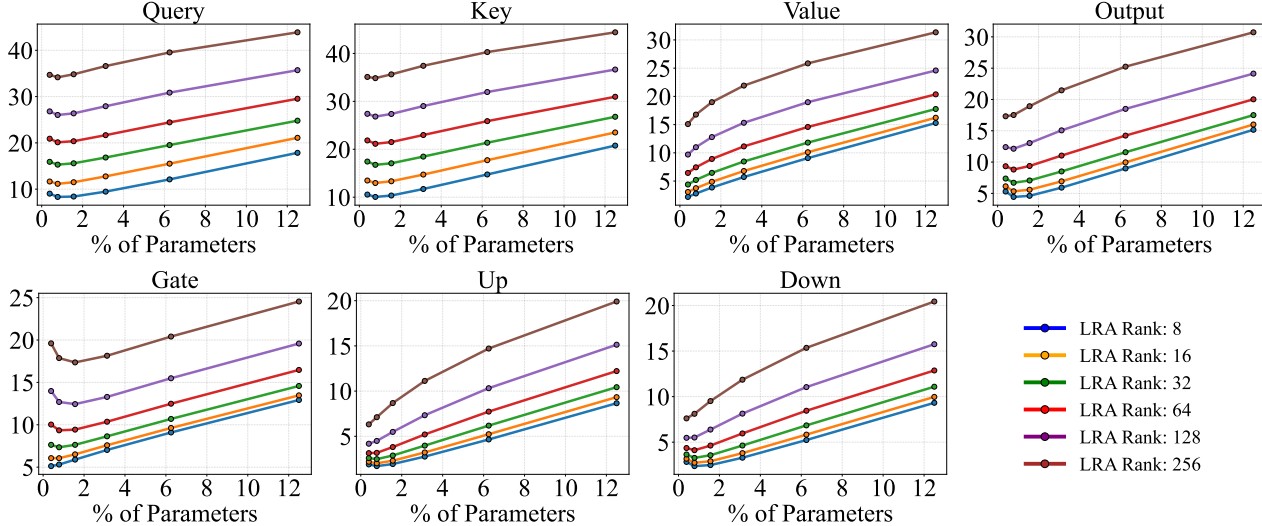

Figure 17: The overlap ratio of parameters selected by `LIFT` and by Weight Magnitude.

$$d = \frac{1}{n} \sum_{i=1}^{n} d_i \tag{8}$$

This should reflect the alignment of a fine-tuned singular vector and the eigensubspace spanned by the pre-trained model. An alignment score of 1 represents that the singular vector aligns perfectly with the pre-trained eigenspectrum, and 0 represents that the singular vector is orthogonal to the pre-trained eigenspectrum.

