# OpenReview forum: "LIFT the Veil for the Truth: Principal Weights Emerge after Rank Reduction for Reasoning-Focused Supervised Fine-Tuning"
_ICML.cc/2025/Conference — ICML 2025 poster_

### Official Review · Reviewer_yeg4 · 2025-03-11

**Overall Recommendation:** 4

**Summary:**

This paper presents LIFT (Low-rank Informed Sparse Fine-Tuning), which introduces the idea of Principal Weights—parameters with the largest magnitude after low-rank approximation—as the most critical ones for LLM fine-tuning. This research in very important to the community, as sparse fine-tuning methods have been largely ignored in the LLM era, even though they were effective for pre-LLM models. The authors argue that one of the reasons for this is that sparse fine-tuning struggles to identify which parameters matter, and they address this with a fundamentally new approach, i.e., using low-rank decomposition to first filter noise and then select key weights based on magnitude.

The method is novel, well-motivated, and well-executed. Sparse fine-tuning has been overlooked in LLMs, and this paper does a fantastic job of showing that it can be revived and improved with low-rank decomposition. The results are strong, the insights are compelling, and the memory efficiency makes it highly practical.

There are some minor things that could be improved (discussion of computational cost, layerwise analysis), but these are not major flaws—more like areas for further exploration. Overall, this is a clear strong accept. It pushes the field forward in an important way, and I expect it will have significant impact in both research and practical applications of LLM fine-tuning.

## update after rebuttal
Authors have addressed my concerns. Thus, I keep it as accept.

**Claims And Evidence:**

The claims in the paper are well supported. The insights that Principal Weights are important to LLMs’ pre-trained knowledge are well supported by their pre-liminary study in Section 4. The effectiveness of LIFT is further supported with extensive experiments in Section 5. I didn’t see how many runs are provided in Section 5. I believe the results will be more significant, if this is the averaged performance over >3 runs. Another claim of LIFT learns better on target domains and preserve better on source domains is also supported in Figure 4.

One issue is that the computational overhead of LIFT isn’t fully explored. While the authors have mentioned that the comparison in their experiments are fair, it is important to report the exact memory costs in the main tables to provide a full picture to audiences.

**Essential References Not Discussed:**

https://arxiv.org/abs/2405.19597

**Experimental Designs Or Analyses:**

Yes, the authors provide strong evidence that this approach revives sparse fine-tuning, making it competitive with (and often better than) Full FT and LoRA-like approaches. One question is whether the results are averaged over multiple runs, given the fact that LLM fine-tuning can be quite noisy. The analysis in this paper is intensive.

Most existing PEFT approaches like LoRA and its variants focus on adding low-rank adapters, they are reported to struggle with large-scale settings (https://arxiv.org/abs/2405.09673). The observation that low-rank decomposition can help sparse fine-tuning by denoising first, then selecting weights, is non-trivial and well-motivated.

**Methods And Evaluation Criteria:**

The idea of Principal Weights itself is new and makes sense. To my best knowledge, denoising first, and then selecting weights for fine-tuning is non-trivial and well-motivated. Previous work of LASER (https://arxiv.org/abs/2312.13558) demonstrate that rank reduction using SVD has a denoising effect on LLMs, enhancing performance, which provides insights to the selection of Principal Weights.

Regarding the evaluation criteria, multiple commonly used benchmarks are evaluated, including commonsense reasoning, math reasoning, and GLUE. The baselines in this paper include SOTA low-rank approaches LoRA, DoRA, and PiSSA, as well as a recent sparse fine-tuning baseline, i.e., S2FT. Some might say that there are many more PEFT approaches over there, but I believe that two most important baselines are already included, (1) PiSSA: SVD-based low-rank adaptation; (2) S2FT: a strong baseline of sparse fine-tuning.

One natural follow-up question to authors is how the rank-level affect the effectivenss of LIFT.

**Other Comments Or Suggestions:**

Some minor typos:
Line 144: contains more -> contain more
Line 250: state-of-the-srt -> state-of-the-art
Line 437: We hope that these problems can lead to -> We hope that these problems can inspire

**Other Strengths And Weaknesses:**

Strengths：I really like the title “LIFT the Veil for the Truth” which subtly incorporates the algorithm name in a meaningful way, so kudos to the authors for that.

Weaknesses：One thing that’s missing is a layerwise breakdown. We know from previous work that FFN layers and Attention layers behave very differently in fine-tuning. Does LIFT work equally well on both? Or are some layers more important than others? A study on layerwise sensitivity would be really interesting and could also help optimize LIFT further.

**Questions For Authors:**

Does LIFT work equally well on both? Or are some layers more important than others? How about depth?
How the rank-level affect the effectivenss of LIFT?
What are the exact memory costs in the main tables to provide a full picture to audiences?
How many random seems are reported in main tables?

**Relation To Broader Scientific Literature:**

LIFT algorithm is highly related to previous literature of using low-rank approximation for LLM denoising (https://arxiv.org/abs/2312.13558, https://arxiv.org/pdf/2406.03068). Moreover, sparse fine-tuning is an active research direction before the popularity of LLMs, where the authors have provided a paragraph of related work.

**Theoretical Claims:**

This paper does not have theoretical claims.

---

> ### Author Rebuttal · Authors · 2025-04-01
>
> Dear reviewer, thank you for the insightful and constructive feedback. We'd like to address your concerns as follows. **The supplementary figures/tables are in the [rebuttal link here](https://github.com/icml12437/ICML2025_12437).**
> # Q1: Layer-wise analysis on LIFT
>
> In Appendix G.4 of our paper, we analyze the **layer-wise effect of LIFT**, where we compare the performance of LIFT in **fine-tuning a single type of layers**. Our results show that fine-tuning only MLP layers yields significantly better results than fine-tuning attention layers. We hypothesize that attention components mostly store information on token relations rather than task-specific knowledge. On the other hand, MLP layers are more adaptive to downstream tasks, and fine-tuning them is more effective.
>
> Based on that insight, we explore the possibility of further improving the efficiency of LIFT by **only fine-tuning MLP layers**. The table below shows the results of LLaMA-2-7B on arithmetic datasets where we only fine-tune the MLP layers (LIFT_MLP), and only fine-tune the attention layers (LIFT_Attn). We can see that LIFT_MLP has similar performance to the full version of LIFT, while LIFT_Attn has drastically worse performance. This suggests that for LIFT, fine-tuning MLP layers is more effective than fine-tuning Attention layers.
>
> ||MultiArith|GSM8K|AddSub|AQuA|SingleEQ|SVAMP|MAWPS|Avg|
> |-|-|-|-|-|-|-|-|-|
> |**LIFT**|98.67|47.31|92.66|**26.77**|**96.85**|**63.6**|**90.34**|**73.74**|
> |**LIFT_MLP**|**99.66**|**47.61**|91.90|25.59|95.67|62.6|**90.34**|73.34|
> |**LIFT_Attn**|95.00|43.75|91.14|25.59|91.73|60.1|86.55|70.55|
> |Full FT|98.17|46.55|**93.67**|22.05|96.85|63.2|89.08|72.79|87.54|
> |LoRA|98.00|47.76|92.41|23.62|95.08|62.9|90.76|72.93|
>
>
>
> # Q2: How does the rank-level affect the effectiveness of LIFT?
>
> In **Appendix G.5 and Fig. 15** of our paper, we study the influence of rank in low-rank approximation (LRA Rank) on the performance of LIFT. We found a correlation between LRA Rank and the number of trainable parameters, that **the optimal LRA Rank increases as we select more parameters to train**. In practice, when LIFT has the same number of parameters as LoRA, we find that **using LRA Rank similar to LoRA rank yields optimal performance**. We note that LRA Rank may differ with different models, using some layer-adaptive metrics could bring further performance gain to LIFT.
>
> # Q3: More discussions on related references
> The SVFT paper [1] introduces a PEFT method called SVFT, which performs SVD on pre-trained weights and fine-tunes the resulting SVD factors. In contrast, our work considers sparse fine-tuning and proposes a novel approach for selecting Principal Weights to update. Furthermore, by storing only the optimizer states associated with these Principal Weights, LIFT significantly reduces memory overhead. We will cite SVFT in the revised version of our paper.
>
> # Q4: Detailed training settings and memory costs
>
> ## Training Settings
>
> In Fig. 3 in our paper, we showed the results of LIFT on GSM8K datasets with 4 random seeds. Due to resource limitations, for other experiments we only report the results of one seed. In Table 8 in the rebuttal link, we present the results on arithemetic datasets with four random seeds (Table 2 in the paper), which further prove the robust performance of LIFT, where it outperforms other baselines under four seeds.
>
> ## Number of Trainable Parameters and GPU Memory Costs
> In all experiments in the paper, we **compare the best results of different methods among a range of parameter sizes**. Specifically, when comparing LIFT with LoRA-like methods, we search the LoRA rank in {16, 32, 64, 128, 256}, and **LIFT with the same parameter counts** to ensure fair comparison, and pick their best results. In practice, we find that the LIFT and LoRA-like methods typically have the best performance at **rank = 128**, similar to results from PiSSA paper [2].
>
> In Fig. 6 of Appendix B, we present the memory breakdown of LIFT compared to Full FT and LoRA under optimal settings. We showed that LIFT achieves similar memory overhead as LoRA, significantly lower than Full FT. Furthermore, we explored only fine-tuning MLP layers with LIFT, which further improves memory advantages over LoRA, while maintaining test performance (details are in the Q2 response to Reviewer ceyi).
>
> # Q5: Typos
>
> We thank the reviewer for pointing out the typo, and we will fix it in the revised version of the paper.
>
> ### References
> [1] Lingam et al, 2024
> [2] Meng et al, 2024

---

> > ### Comment · Reviewer_yeg4 · 2025-04-07
> >
> > Thank you for the authors’ responses, which have addressed my concerns. I keep it as accept.

---

> > > ### Author Response · Authors · 2025-04-07
> > >
> > > Dear Reviewer yeg4,
> > >
> > > We sincerely thank you for your positive response. We will include the additional experimental results and texts in the revised version of our paper.
> > >
> > > Best,
> > > Authors

---

### Official Review · Reviewer_AcQs · 2025-03-11

**Overall Recommendation:** 5

**Summary:**

The authors propose a novel sparse fine-tuning approach, LIFT, which identifies so-called Principal Weights. By only training these Principal Weights, LIFT outperforms full-parameter fine-tuning in multiple benchmarks including commonsense reasoning, math reasoning, and GLUE tasks. Specifically, Principal Weights are the weights that have the largest magnitude after performing low-rank reduction.

**Claims And Evidence:**

The paper introduces the Low-rank Informed Sparse Fine-Tuning (LIFT) method and provides substantial evidence supporting its effectiveness. The claim of LIFT's superior performance is validated through extensive experimental evaluation across three distinct tasks: (1) Commonsense Reasoning, (2) Arithmetic Reasoning, and (3) Natural Language Understanding. Additionally, the paper robustly supports the claim that LIFT identifies Principal Weights through experiments in Figure 2.

**Essential References Not Discussed:**

https://arxiv.org/pdf/2401.16405 this is very early sparse finetuning methods for LLMs, which is missing in this paper.

**Experimental Designs Or Analyses:**

(1)Good evaluation, comparing against a number of alternative state of the art models. I appreciate the fact that all experiments are run with four random seeds, which demonstrate the performance gain of LIFT is significant.

(2)It would be helpful if the paper explicitly analyzed the ratio or overlap between Principal Weights and original largest-magnitude weights across different layers. Understanding whether Principal Weights significantly differ from massive weights, and how this ratio varies across model layers, could offer deeper insights into LIFT’s effectiveness and behavior.

**Methods And Evaluation Criteria:**

The motivation for identifying principal weights via low-rank decomposition is clearly justified by Figure 2. The effectiveness of LIFT is thoroughly demonstrated through extensive experiments across diverse tasks and varying model sizes.

**Other Comments Or Suggestions:**

Typos: state-of-the-srt -> state-of-the-art in line 250.

**Other Strengths And Weaknesses:**

Strengths:
Novel idea nicely explained and motivated. It is interesting to see that crucial weights emerge after performing SVD.
LIFT demonstrates stronger generalization, balancing the learning of new task-specific knowledge with minimal forgetting of the source domain knowledge.

Weakness:
The paper's identification of Principal Weights is insightful; however, recent research (e.g., the Massive Weights in LLMs paper) suggests original large-magnitude weights are also crucial. It might be beneficial to explore combining both Principal Weights and original Massive Weights to further enhance performance, rather than relying exclusively on one set.

**Questions For Authors:**

Considering that unstructured sparse fine-tuning is not well supported by GPUs, whereas structured sparse fine-tuning enjoys efficient GPU support, Can this LIFT technique be adapted to structured sparse finetuning?

**Relation To Broader Scientific Literature:**

The low rank finds principle weights is related to the paper “ From Galore to Welore: how low-rank weights no-uniformly emerge from low-rank gradients” (https://arxiv.org/abs/2407.11239.)

**Theoretical Claims:**

No theoretical claims.

---

> ### Author Rebuttal · Authors · 2025-04-01
>
> Dear reviewer, thank you for the insightful and constructive feedback. We'd like to address your concerns as follows.
>
> # Q1: Overlap between Principal Weights and the original largest-magnitude weights
>
> In **Appendix G.6** of our paper, we discussed the overlap between parameters selected by LIFT and parameters selected by weight magnitude. In Fig. 16, we plotted the overlap ratio between LIFT and weight magnitude on different layer types as the number of trainable parameters varies. We showed that the overlap is generally below 20%, while different layers have different overlap ratios. We note that the overlap on MLP layers are significantly lower than that on attention layers. This suggests that the low-rank approximation of the Query and Key matrices are close to the original matrix, and Query and Key modules are low-rank in nature.
>
> The above results indicate that the *Principal Weights* of LIFT and original large-magnitude weights do not overlap heavily, and how to combine the two types of weights is indeed a promising direction of future work.
>
> # Q2: Missing discussions on recent sparse fine-tuning paper.
>
> Here we discuss the paper "Scaling Sparse Fine-Tuning to Large Language Models" [1]. This paper proposed SpIEL approach to reduce the need for computing all gradients during backward propagation, which scales the gradient memory linearly with the selected weights. While our method LIFT proposed a novel way to select  Principal Weights for sparse fine-tuning and store only the optimizer states of Principal Weights to significantly reduce the memory overhead.
>
> **We also compare LIFT with SpIEL on the GSM8K dataset.** We use the training setting as in Fig. 3 in our paper. For both methods, we searched learning rate among {5e-5, 1e-4, 2e-4, 5e-4} and trainable parameters corresponding to LoRA rank among {16, 32, 64, 128, 256} to obtain the best results. The table below shows that LIFT significantly outperforms SpIEL with both LLaMA-2-7B and LLaMA-3.2-3B model.
>
> |**GSM8K**|LIFT|SpIEL|Full FT|
> |-|-|-|-|
> |LLaMA-3.2-3B|**46.46**|43.76|44.50|
> |LLaMA-2-7B|**24.24**|21.61|22.57|
>
> What's more, combining LIFT with the way SpIEL reduces memory overhead to further pursue memory efficiency is a promising direction for future work.
>
>
> # Q3: Can LIFT technique be adapted to structured sparse finetuning?
>
> To evaluate whether LIFT can work in a structured fine-tuning fashion, we perform initial experiments on structured block-sparsity, where we select a number of $4\times4$ blocks to fine-tune (**LIFT_Structured**). We select 5% of all parameters to train (corresponding to LoRA rank $\approx$ 128). We choose to fine-tune LLaMA-2-7B model with MATH-10K dataset and evaluate on seven arithmetic reasoning tasks. The results are as follows.
>
> ||MultiArith|GSM8K|AddSub|AQuA|SingleEQ|SVAMP|MAWPS|Avg.|
> |-|-|-|-|-|-|-|-|-|
> |**LIFT_Structured**|98.33|**48.07**|93.16|25.98|95.47|**65.1**|89.92|**73.72**|
> |**LIFT**|**98.67**|47.31|92.66|**26.77**|**96.85**|63.6|**90.34**|**73.74**|
> |Full FT|98.17|46.55|**93.67**|22.05|96.85|63.2|89.08|72.79|87.54|
>
> From the table above, we can see that LIFT still achieves great performance under structured sparsity. LIFT_Structured has almost the same performance as original LIFT, while outperforming Full FT. This suggests that LIFT has the potential to be adapted to structured, sparse fine-tuning to achieve further computation acceleration.
>
> # Q4: Typos
>
> We thank the reviewer for pointing out the typo, and we will fix it in the revised version of the paper.
>
> ### References
> [1] Ansell et al, 2024

---

> > ### Comment · Reviewer_AcQs · 2025-04-07
> >
> > I thank the authors for the detailed responses, which address my concerns.  Considering the significance of the work in PEFT domain, especially for sparse-based finetuning methods, I would like to raise my recommendation.

---

> > > ### Author Response · Authors · 2025-04-07
> > >
> > > Dear Reviewer AcQs,
> > >
> > > We sincerely appreciate your positive feedback and your decision to raise your recommendation. We will include the additional experimental results and texts in our revised draft.
> > >
> > > Best,
> > > Authors

---

### Official Review · Reviewer_ceyi · 2025-03-18

**Overall Recommendation:** 3

**Summary:**

This paper proposes a novel sparse fine-tuning method, LIFT, that identifies and fine-tunes the critical parameters (Principal Weights) through SVD-based rank reduction. Extensive experiments demonstrate that LIFT significantly outperforms existing parameter-efficient fine-tuning (PEFT) approaches, including LoRA, as well as full fine-tuning. The paper also provides valuable insights through thorough analyses, including eigenspace and eigenspectrum investigations and detailed ablation studies.

**Claims And Evidence:**

The claims presented in the paper are well-supported by comprehensive empirical results across various tasks, demonstrating consistent effectiveness.

**Essential References Not Discussed:**

The paper adequately discusses relevant literature but omits discussions and comparisons with several recent sparse fine-tuning methods, including:

[r1] Sparse is Enough in Fine-tuning Pre-trained Large Language Models, ICML 2024

[r2] SMT: Fine-Tuning Large Language Models with Sparse Matrices, ICLR 2025

[r3] Scaling Sparse Fine-Tuning to Large Language Models, arXiv preprint arXiv:2401.16405, 2024.

**Experimental Designs Or Analyses:**

The experimental setup is clearly structured and adequately covers a broad spectrum of tasks. However, the comparison baselines are primarily related to LoRA-based methods, while Sparse FT methods are less extensively compared, with only S2FT included as a baseline.

**Methods And Evaluation Criteria:**

The benchmarks selected in this paper are suitable for evaluating the proposed method's performance.

**Other Comments Or Suggestions:**

- In the analysis of eigenspace and eigenspectrum, incorporating metrics like effective rank (as presented in Figure 8 of HiRA [r4]) might better represent the full singular value spectrum.
- Expand the method comparison by including recent sparse fine-tuning methods as well as advanced high-rank fine-tuning techniques, like HiRA, which could further demonstrate LIFT’s strengths.

[r4] HiRA: Parameter-Efficient Hadamard High-Rank Adaptation for Large Language Models, ICLR 2025

**Other Strengths And Weaknesses:**

Strengths:
- The paper is clearly structured, well-written, and easy to follow.
- The proposed method (LIFT) is conceptually sound, with thorough analyses and ablation studies to support its effectiveness.
- LIFT achieves superior performance compared to Full FT, while maintaining memory efficiency similar to LoRA and other parameter-efficient methods.

Weaknesses:

- The tables presented in the paper do not explicitly report the actual GPU memory consumption or the exact number of trainable parameters for each method. Additionally, details such as the specific rank setting for LoRA are unclear; from Figure 11, it seems the LoRA rank is at least 512, significantly different from the common settings (typically 8-32 for LLaMA), potentially leading to unfair comparisons. It is advisable to supplement this information explicitly.

- According to Appendix B, LIFT still has higher GPU memory usage during training compared to LoRA and its variants. The part of GPU memory consumed by activations can be significantly reduced using checkpointing techniques, which may make LoRA’s memory efficiency advantage even clearer.

**Questions For Authors:**

Q1: The paper states that LIFT balances learning and forgetting effectively, yet from Figure 5, the weights undergo significant changes, especially Principal Weights. Could such substantial modifications negatively impact the performance on Out-of-Distribution (OOD) tasks to some extent?

Q2: Could the authors add loss convergence curves for each method in the main experiments, similar to Figure 12, to further clarify the training behavior and convergence speed?

**Relation To Broader Scientific Literature:**

The paper introduces a straightforward yet highly effective approach for sparse FT. Additionally, it explores multiple parameter selection strategies, providing valuable insights and analysis.

**Theoretical Claims:**

N/A

---

> ### Author Rebuttal · Authors · 2025-04-01
>
> Dear reviewer, thank you for the constructive feedback. We'd like to address your concerns as follows. **The supplementary figures/tables are in the [rebuttal link here](https://github.com/icml12437/ICML2025_12437).**
> # Q1: Comparison with recent sparse fine-tuning methods
> We compare LIFT with sparse fine-tuning methods, SIFT [1] and SpIEL [2] (since SMT is not opensourced, we compare with other two). We will cite these papers in the revised version.
>
> **First, we compare LIFT with SIFT on GLUE tasks** following the SIFT paper, with the same number of trainable parameters (5% total parameters). We use the RoBERTa-large model and search LR of SIFT and LIFT in {5e-5, 7e-5, 1e-4, 2e-4} and compare their best results. The table below shows that LIFT outperforms SIFT on all GLUE tasks, while outperforming Full FT on almost all tasks.
> ||MNLI|SST2|MRPC|CoLA|QNLI|QQP|RTE|STSB|Avg.|
> |-|-|-|-|-|-|-|-|-|-|
> |**LIFT**|**90.79**|96.67|90.93|**70.44**|**94.69**|**92.38**|**87.00**|**92.58**|**89.44**|
> |SIFT|89.91|**96.79**|89.95|66.29|93.04|88.49|87.00|92.27|87.97|
> |Full FT|90.58|96.22|**91.91**|68.55|94.47|91.52|85.92|92.21|88.92|
>
> **Second, we compare LIFT with SpIEL on GSM8K** (See Q2 response to Reviewer AcQs). We show that LIFT significantly outperforms SpIEL on both LLaMA-2-7B and LLaMA-3.2-3B models.
> # Q2: Number of trainable parameters and GPU memory cost of LIFT
> ## 2.1. Number of trainable parameters
> In all experiments in the paper, we **compare the best results of different methods in a range of parameter sizes**. When comparing LIFT with LoRA-like methods, we search the LoRA rank in {16, 32, 64, 128, 256}, and **LIFT with the same parameter counts** to ensure fair comparison. We find that LIFT and LoRA-like methods typically perform best at **rank = 128**.
>
> In addition, in our analytical results (e.g. Fig. 11), we compare LIFT and LoRA both with **rank = 128**. Some layers of LoRA have rank larger than 128 is likely due to numerical errors when computing the rank.
>
> ## 2.2. GPU memory cost of LIFT
> We note that although the memory cost of LIFT (as in Fig. 6 in Appendix B) is slightly larger than LoRA, we can further reduce the memory cost of LIFT while preserving performance.
>
> In Appendix G.4, we showed that fine-tuning MLP layers is more effective than fine-tuning attention layers. The table below shows the results of LLaMA-2-7B on arithmetic datasets where we **only fine-tune the MLP layers (LIFT_MLP)**. We see that LIFT_MLP has similar performance to LIFT. Furthermore, only fine-tuning MLP layers further reduces the memory usage on gradients and optimizer states. In Fig. 17 of the rebuttal link, we see that LIFT_MLP achieves better memory efficiency than LoRA under optimal settings.
> ||MultiArith|GSM8K|AddSub|AQuA|SingleEQ|SVAMP|MAWPS|Avg.|
> |-|-|-|-|-|-|-|-|-|
> |**LIFT**|98.67|47.31|92.66|**26.77**|**96.85**|**63.6**|**90.34**| **73.74**|
> |**LIFT_MLP**|**99.66**|**47.61**|91.90|25.59|95.67|62.6|**90.34**| **73.34**|
> |Full FT|98.17|46.55|**93.67**|22.05|96.85|63.2|89.08|72.79|87.54|
> |LoRA|98.00|47.76|92.41|23.62|95.08|62.9|90.76|72.93|
> # Q3: Could LIFT negatively impact the performance on OOD tasks?
> We believe although model weights undergo significant changes during fine-tuning, because only a small set of parameters are changed while most weights remain unchanged (see the center of histogram in Fig. 5), the model retains its fundamental capacities that enable it to generalize to OOD settings.
>
> In Sec. 7.1, we also analyzed the generalization performance of LIFT by training models with arithmetic datasets and evaluating on commonsense tasks (OOD). This study verifies that LIFT achieves stronger OOD performance compared to other baselines.
> # Q4: Metrics like effective rank
> Here we compare LIFT with HiRA [3] and evaluate the effective rank of other baselines.
>
> The table below shows the results of LIFT and HiRA on arithmetic datasets (same as Table 2 in our paper). We search HiRA rank in {16, 32, 64, 128, 256, 512} and found that HiRA performs best at rank = 512. We see that LIFT (rank = 128) outperforms HiRA with rank = 512.
> ||MultiArith|GSM8K|AddSub|AQuA|SingleEQ|SVAMP|MAWPS|Avg.|
> |-|-|-|-|-|-|-|-|-|
> |**LIFT**|**98.67**|**47.31**|**92.66**|**26.77**|**96.85**|**63.6**| **90.34**|**73.74**|
> |HiRA|98.50|46.70|91.65|25.59|95.67|61.5|89.50|72.72|
>
> In Fig. 19 of the rebuttal link, we show the effective rank of different methods. We see that the update matrix of LIFT has larger effective rank than LoRA and PiSSA, close to Full FT, while slightly lower than HiRA. We believe although effective rank is a good indicator of model capacity, it's insufficient to soly rely on it to predict performance.
> # Q5: Training curve of LIFT
> In Fig. 18 of the rebuttal link, we show the training loss curve of all methods in Table 2 results. We see that the convergence speed of LIFT is on par with Full FT, notably faster than other PEFT methods.
> ## References
> [1] Song et al, 2024
> [2] Ansell et al, 2024
> [3] Huang et al, 2025

---

> > ### Comment · Reviewer_ceyi · 2025-04-03
> >
> > Thank you for the reply. While some of my concerns have been addressed, several key issues remain unresolved:
> >
> > #### Explanation of Numerical Errors
> >
> > > The claim that ranks exceeding 128 are due to “numerical errors” is vague and unconvincing. Please specify what numerical instability would cause a computed rank to be higher than the intended value.
> >
> > #### Discrepancy in Analytical Results
> >
> > > Figures 11 and 19 show that many LoRA components and layers exhibit near-zero ranks, which appears inconsistent with the stated use of **rank = 128**. This discrepancy raises concerns about whether LoRA was fully utilized during training.
> >
> > #### Lack of Parameter Search Details
> >
> > > You state that LoRA ranks in {16, 32, 64, 128, 256} were explored, yet the paper does not provide any evidence of this search process. In particular, no table explicitly reports the rank settings used. For transparency and fair comparison, it is important to **include performance results for each method across different ranks**.
> >
> > I encourage the authors to provide these missing details to substantiate their claims. **If these issues remain unresolved, I will need to reconsider my overall assessment of the paper.**

---

> > > ### Author Response · Authors · 2025-04-05
> > >
> > > Dear Reviewer ceyi,
> > >
> > > Thank you for your thoughtful follow-up. We acknowledge that these concerns are important for substantiating our claims. We now address your points more thoroughly below. Updated figures and tables can be found at **[rebuttal link](https://github.com/icml12437/ICML2025_12437)**.
> > >
> > > ## 1. Discrepancy in Analytical Results.
> > >
> > > First, we thank the reviewer for the insightful observation! We would like to clarify that Figures 11 and 19 originally presented results using incorrect LoRA checkpoints (with rank = 16 instead of the intended rank = 128). We apologize for this oversight. The corrected results are now provided in Figures 19 and 21 in the rebuttal link. Additionally, we have included the averaged rank and effective rank across different layer types in Tables 9 and 10 of the rebuttal link.
> > >
> > > **We can see that the overall claim still holds:** LIFT with sparse fine-tuning retains a substantially higher rank and effective rank than LoRA.
> > >
> > > ## 2. Explanation of Numerical Rank Calculation.
> > >
> > > **LoRA’s computed rank is higher than LoRA’s target rank due to the default threshold parameter in `torch.linalg.matrix_rank` function being too low. We use a more robust threshold in our updated results.**
> > >
> > > The built-in `torch.linalg.matrix_rank` function computes the rank  by counting singular values greater than a threshold $\tau$, which has the default value:
> > > $$\tau = \max{(m, n)} \times \sigma_{\max} \times \epsilon,$$
> > > where $(m, n)$ is the matrix shape, $\sigma_{\max}$ is the largest singular value, and $\epsilon$ is precision of input data type. **If $\tau$ is lower than the rounding error commited during LoRA’s update matrix evaluation, the computed rank may exceed LoRA's target rank.**
> > >
> > > In our experiments, the update matrices were evaluated by subtracting the base weights from the fine-tuned weights with LoRA adapters merged. When we computed the rank in Fig. 11 using `torch.linalg.matrix_rank`, we used the default threshold which was too low to guarantee that the LoRA’s computed rank does not exceed the LoRA’s target rank.
> > >
> > > **For further investigation, we raise the threshold in `torch.linalg.matrix_rank`.**  Fig. 20 in the rebuttal link shows the computed ranks of LoRA updates (with Rank = 16) of Query matrices under varying threshold-raising factors. As the threshold increases (e.g., by a factor of 10), LoRA’s numerical rank begins to align with its target rank and we no longer see high values over 500.
> > >
> > > Therefore to obtain robust rank comparison, in our updated results (Fig. 21 and Table 9), we compute the ranks of LIFT, Full FT and LoRA update matrices with a **threshold set to 10 times the default**. We observed the same overall trend as before: LIFT consistently achieves a significantly higher computed rank than LoRA.
> > >
> > > ## 3. Lack of Parameter Search Details
> > > Here we report the test results under different ranks for all methods in our main experiments. Note that since LIFT focuses on improving the best performance, in the main tables of the paper, **we report the best results among all the ranks for each method** (highlighted in the tables below). Due to space limit, here we present the rank search results from Table 2 in the paper. We will incorporate the search table for more experiments in our camera-ready version. All baseline results that were reproduced are similar to previously reported results such as S2FT paper.
> > >
> > > In the below tables we show the **mean accuracy on 7 arithmetic reasoning tasks (as in Table 2 in our paper)**. We can see that for LLaMA-2-7B and LLaMA-3.2-3B, LIFT and PEFT methods generally achieve the best performance at Rank = 128, under which LIFT significantly outperforms other baselines. When the LoRA rank is low (e.g., 16), all sparse fine-tuning methods (such as LIFT and S2FT) exhibit degraded performance. This is likely because sparse fine-tuning updates only a tiny, insufficient subset of parameters, whereas other PEFT approaches like LoRA modify the entire weight matrix, allowing for more expressive adaptation even at lower ranks.
> > >
> > > |LLaMA-2-7B|16|32|64|128|256|
> > > |-|-|-|-|-|-|
> > > |**LIFT**|70.91|71.09|72.74|**73.74**|73.67|
> > > |Full FT|72.79|72.79|72.79|72.79|72.79|
> > > |S2FT|67.78|71.78|72.48|**73.10**|72.63|
> > > |PiSSA|71.57|71.82|72.54|**73.03**|72.54|
> > > |DoRA|71.10|71.74|**72.42**|71.83|71.81|
> > > |LoRA|70.91|71.74|72.81|**72.93**|72.24|
> > >
> > > |LLaMA-3.2-3B|16|32|64|128|256|
> > > |-|-|-|-|-|-|
> > > |**LIFT**|74.51|76.06|76.61|**77.06**|76.41|
> > > |Full FT|76.45|76.45|76.45|76.45|76.45|
> > > |S2FT|72.17|74.86|75.12|**75.58**|75.20|
> > > |PiSSA|75.26|74.97|75.55|**75.69**|72.65|
> > > |DoRA|75.19|75.00|75.16|**75.59**|75.57|
> > > |LoRA|74.82|75.59|75.66|**75.71**|75.64|
> > >
> > > We hope these results address your concerns, and we appreciate your continued feedback. These results will be incorporated in the camera-ready manuscript. Thank you for helping us improve our work.

---

### Official Review · Reviewer_kFAr · 2025-03-24

**Overall Recommendation:** 2

**Summary:**

This paper introduces a sparse fine-tuning approach, LIFT, which identifies and updates what the authors call “Principal Weights” in LLMs. The central claim is that the most critical parameters for downstream fine-tuning can be found by first applying low-rank approximation (e.g., SVD) to each weight matrix, then selecting the largest-magnitude weights from the approximated matrix. These selected parameters are subsequently updated during training, while the rest remain frozen. The paper reports that LIFT achieves performance on par with or exceeding full fine-tuning across various tasks (e.g., arithmetic and commonsense reasoning), all while using a memory footprint comparable to parameter-efficient fine-tuning (PEFT) methods like LoRA.

======

After reviewing the full discussion again, I find that two issues still remain unresolved and, in my opinion, are unconvincing:

- The authors mention that they follow the experimental setup of WeLore, yet they do not include WeLore in their comparisons or discussions in the manuscript.

- The implementation of the method directly modifies original model weights, causing position changes during training that violate sparsity constraints. Thus, the approach aligns more with memory-efficient or sparse training methods rather than PEFT.

For the reasons stated above, I maintain my current rating.

**Claims And Evidence:**

The paper’s main claims—that “Principal Weights” identified via low-rank approximation are the most important for fine-tuning and that the resulting LIFT method outperforms both full fine-tuning and prior PEFT approaches—feel somewhat overstated. While the empirical results are suggestive, the authors don’t convincingly prove that this exact selection strategy is uniquely optimal.

**Essential References Not Discussed:**

They mention some sparse and low-rank methods (like LoRA), but skip other key variants on dynamic or structured pruning (e.g., top-k fine-tuning approaches) that also use magnitude- or gradient-based selection.

**Experimental Designs Or Analyses:**

Their experimental setup mostly follows standard fine-tuning procedures, but there’s little discussion of hyperparameter tuning or random seed variability. It’s unclear how stable these results are across multiple runs.

**Methods And Evaluation Criteria:**

They use typical LLM benchmarks—like arithmetic and commonsense tests—to judge performance. While these tasks do show some variety, there’s no deeper or more specialized evaluation that might reveal the approach’s limits or weaknesses. The metrics are standard accuracy-based scores, which is fine, but there’s little beyond that (e.g., no real-world or higher-level analysis).

**Other Comments Or Suggestions:**

A clearer, more concise discussion of hyperparameters for each experiment and how they’re tuned would help readers reproduce results.

**Other Strengths And Weaknesses:**

A notable plus is that the paper lays out an easy-to-replicate method and backs it up with multiple experiments. The authors do a decent job packaging a simple rank-then-sparsify idea into a seemingly effective pipeline. However, the novelty is fairly modest. The paper also sometimes feels repetitive when explaining the approach, and the clarity on hyperparameters and training details could be improved.

**Questions For Authors:**

You focus on arithmetic and commonsense tasks—have you tested more diverse domains (e.g., structured QA, code generation)? If so, how does LIFT fare there, especially on tasks known for high domain shift or specialized knowledge?

**Relation To Broader Scientific Literature:**

They do reference recent efforts like LoRA, low-rank approximation approaches, and sparse-adapter methods. However, the paper mostly leans on known results about low-rank structures in large language models (e.g., Eckart–Young–Mirsky) and doesn’t deeply compare against or incorporate broader theoretical and empirical work on compressive adaptation (like structured pruning, block-wise sparsity, etc.).

**Theoretical Claims:**

There isn’t much formal proof offered

---

> ### Author Rebuttal · Authors · 2025-04-01
>
> Dear reviewer, thank you for the constructive feedback. We'd like to address your concerns as follows. **Please find the supplementary figures/tables in the [rebuttal link here](https://github.com/icml12437/ICML2025_12437).**
> # Q1: Higher-level analysis on LIFT
> In our paper, we conduct higher-level analysis on LIFT through empirical analysis and eigenspectrum analysis.
>
> **First, we show that parameters selected by LIFT are crucial to model performance.** In Sec. 4 and Fig. 2, we show that LIFT selects principal weights that are sensitive to random perturbations, which has significant impact on the model performance, thereby validating that these weights are good candidates for fine-tuning.
>
> **Second, we also provided in-depth analysis of the eigenspace of LIFT.** Sec. 7.2 and 7.3 show that LIFT induces larger weight changes (Fig. 5), and significantly increases the rank (and effective rank, see the Q4 response to Reviewer ceyi) of weight updates (Fig. 11), and exhibits greater singular subspace divergence (Fig. 10), all of these demonstrates LIFT shows a greater learning capacity in fine-tuning. Additionally, spectral norm analysis suggests LIFT enhances generalization (Fig. 8, 9), where random perturbations sharply increase the spectral norm in both random matrix and LLM settings, and LIFT's lower gradient norm curves (Fig. 12) further demonstrate its effectiveness, as supported by PiSSA [1].
> # Q2: Evaluation on diverse domains
> We conduct more experiments on QA and Code Generation datasets. These experiments further showcase the effectiveness of LIFT on more diverse domains.
>
> For QA, we use the experimental setup of StrategyQA dataset following recent WeLore paper [2]. For PEFT methods, we consider ranks {16, 32, 64, 128, 256}; for LIFT, we use the same counts of trainable parameters. The learning rates are {1e-5, 2e-5, 5e-5, 1e-4} for Full FT and {5e-5, 1e-4, 2e-4, 5e-4} for others. We select the best-performing config for each method and report the results below. We can see that LIFT achieves notable performance gains than all other methods on both LLaMA-2-7B and LLaMA-3-8B.
> |StrategyQA|LIFT|Full FT|LoRA|DoRA|PiSSA|
> |-|-|-|-|-|-|
> |LLaMA-2-7B|**72.53**|70.61|71.78|71.98|71.26|
> |LLaMA-3-8B|**75.85**|74.81|74.44|74.27|75.19|
>
> For code generation, we adopt the settings from the recent SIFT paper [3], where we fine-tune LLaMA-2-7B with Alpaca dataset for one epoch, and evaluate on the **Humaneval** dataset. From the table below we can see that LIFT outperforms all other methods in both pass@1 and pass@10 settings.
> |**Humaneval**|LIFT|Full FT|SIFT|LoRA|DoRA|
> |-|-|-|-|-|-|
> |Pass@1|**16.46**|15.24|14.02 |13.66|13.96|
> |Pass@10|**31.10**|28.05|30.48|27.44|29.88|
>
> # Q3: Is the results of LIFT robust under multiple seeds?
> In Fig. 3 in our paper, we show the results of LIFT on GSM8K datasets with 4 random seeds. In table 8 in the rebuttal link, we present the results on arithemetic datasets with four random seeds (same as Table 2 in the paper), which further prove that LIFT has robust performance, where it outperforms other baselines under four seeds.
>
> # Q4: Comparison with dynamic/structured pruning methods
> In Fig. 3, we have compared LIFT with a wide range of other dynamic sparse fine-tuning methods, including Top-k Magnitude, Top-k Gradient, and Movement, and showed that LIFT outperforms all other sparse fine-tuning methods. We note that unstructured sparse fine-tuning usually serves as the performance ceiling of structured and block-wise sparsity. In addition, in our main results, we have compared LIFT with S2FT, which is the state-of-the-art structured fine-tuning method by far. These results suggest that LIFT is indeed superior to other common sparse selection methods.
>
> # Q5: Hyperparameter Tuning
> Here we explain the hyperparameter tuning for experiments. We will incorporate detailed information in the revised version of our paper.
> ## Training Hyperparameters
> In all experiments in the paper, we **compare the best results of different methods among a range of parameter sizes**. Specifically, when comparing LIFT with LoRA-like methods, we search the LoRA rank in {16, 32, 64, 128, 256}, and **LIFT with the same parameter counts** to ensure fair comparison, and pick their best results. We find that LIFT and LoRA-like methods typically have the best performance at **rank = 128**, similar to results from PiSSA paper [1].
> ## LIFT Hyperparameters
> The main hyperparameter of LIFT is the **LRA Rank**, which **controls the rank for low-rank approximation** of weight matrices. In **Appendix G.5 and Fig. 15** of our paper, we study the influence of LRA Rank on the performance of LIFT. We found that the LRA Rank of best performance increases as we select more parameters to train. In practice, when we tune LIFT with the same number of parameters as LoRA, we find that **using LRA Rank similar to LoRA rank yields optimal performance**.
> ## References
> [1] Meng et al, 2024
> [2] Jaiswal et al, 2024
> [3] Song et al, 2024

---

> > ### Comment · Reviewer_kFAr · 2025-04-02
> >
> > Thanks for sharing these new experimental results. Overall, I’m feeling better about the paper, but I still see limited novelty here. For now, I’m not inclined to raise my score, though I might change my mind if the AC-led discussion brings up fresh insights.

---

> > > ### Author Response · Authors · 2025-04-02
> > >
> > > Dear Reviewer kFAr,
> > >
> > > Thank you again for your follow-up. To further address your concerns, we'd like to summarize the novelty of our work as follows.
> > >
> > > **First, we propose a novel concept of principal weights which is crucial to LLM fine-tuning, and designed a new sparse fine-tuning algorithm.** We showed that incorporating eigenspectrum and low-rank information can identify principal weights crucial to model performance. To our knowledge, our approach is the first to leverage eigenspectrum decomposition and rank reduction for sparse fine-tuning of LLMs. In terms of the novelty of LIFT algorithm, we received support from the other three reviewers (e.g. "The method is novel, well-motivated, and well-executed." from Reviewer `yeg4`, "Novel idea nicely explained and motivated." from Reviewer `AcQs`)
> > >
> > > **Second, we conduct comprehensive experiments across a wide range of domains**, to show the effectiveness and robustness of our method compared to other state-of-the-art algorithms. We also conduct **in-depth empirical and eigenspectrum analysis** to study the reason behind our method's success. Our experimental setup has also been acclaimed by other reviewers (e.g. "Adequately covers a broad spectrum of tasks." from Reviewer `ceyi`)
> > >
> > > **Third, our work is meaningful to the research community on sparsity and fine-tuning.** Our method showed that sparse fine-tuning can achieve strong performance on LLM fine-tuning. This is meaningful to future research in the community on sparse fine-tuning, as it has fallen behind in the modern LLM era. This significance of our work is also recognized by other reviewers (e.g. "This research is very important to the community", "It pushes the field forward in an important way" from Reviewer `yeg4`).
> > >
> > > In addition, during the rebuttal, we firmly believe that we have addressed all of your concerns, including 1) higher-level analysis of our LIFT method, 2) more evaluation on diverse domains, 3) performance robustness under random seeds.
> > >
> > > We sincerely appreciate your constructive feedback, and we hope you can reconsider your evaluation of our work.
> > >
> > > Best,
> > > Authors

---

### Decision · Program_Chairs · 2025-05-01

**Decision:**

Accept (poster)

**Comment:**

This paper received mixed reviews. After discussion, it was agreed that this paper is  above the ICML acceptance bar.
The authors are encouraged to address the comments of the reviewers to improve this work.
The author’s rebuttal has been carefully read and discussed. The author’s message has been carefully read and considered.